# Query-Guided Spatial–Temporal–Frequency Interaction for Music Audio–Visual Question Answering

**Kun Li**[1,3][*]    **Michael Ying Yang**[2]    **Sami Sebastian Brandt**[3,][†]
[1] University of Twente, NL    [2] University of Bath, UK    [3] IT University of Copenhagen, DK
`k.li@utwente.nl, myy35@bath.ac.uk, sambr@itu.dk`

## Abstract

Audio–Visual Question Answering (AVQA) is a challenging multimodal task that requires jointly reasoning over audio, visual, and textual information in a given video to answer natural language questions. Inspired by recent advances in Video QA, many existing AVQA approaches primarily focus on visual information processing, leveraging pre-trained models to extract object-level and motion-level representations. However, in those methods, the audio input is primarily treated as complementary to video analysis, and the textual question information contributes minimally to audio–visual understanding, as it is typically integrated only in the final stages of reasoning. To address these limitations, we propose a novel **Q**uery-guided **S**patial–**T**emporal–**F**requency (**QSTar**) interaction method, which effectively incorporates question-guided clues and exploits the distinctive frequency-domain characteristics of audio signals, alongside spatial and temporal perception, to enhance audio–visual understanding. Furthermore, we introduce a Query Context Reasoning (QCR) block inspired by prompting, which guides the model to focus more precisely on semantically relevant audio and visual features. Extensive experiments conducted on two AVQA benchmarks demonstrate the effectiveness of our proposed method, achieving significant performance improvements over existing Audio QA, Visual QA, Video QA, and AVQA approaches. The code is released under `https://github.com/lik1996/QSTar`.

## 1 Introduction

Understanding audio–visual scenes in videos is crucial for a wide range of real-world applications, including autonomous driving (Wang et al., 2024), human–computer interaction (Li et al., 2023c), and event localization (Grumiaux et al., 2022). Querying specific information from videos provides an intuitive and interactive interface that enables humans to engage with machines and gain deeper insights into the physical world. As a representative task in multimodal video understanding, Audio–Visual Question Answering (AVQA) (Yun et al., 2021; Yang et al., 2022) requires models to jointly interpret and reason over both aural and visual modalities to answer natural language questions about video content. Unlike traditional Visual or Video QA tasks (Yu et al., 2019; Le et al., 2020) that rely primarily on visual cues, AVQA demands a deeper integration of sound cues and visual content to capture the full semantics of a scene.

In many real-world scenarios, audio conveys critical information that visuals alone may fail to capture. Tasks such as identifying a speaker in a conversation or distinguishing between visually similar events often rely heavily on auditory cues (He et al., 1999; Ji et al., 2021). Therefore, effective AVQA systems must not only interpret visual content but also recognize sound events, understand their temporal dynamics, and model their interactions with visual signals. To facilitate research in this direction, Li et al. (2022) proposed a large-scale dataset for music scene understanding, called MUSIC-AVQA, which has become a standard benchmark for questioning audio–visual content. Similarly, in music AVQA, audio is especially helpful when visual cues are limited (*e.g.,* subtle flute

---

[*]Work done while at IT University of Copenhagen
[†]Corresponding author

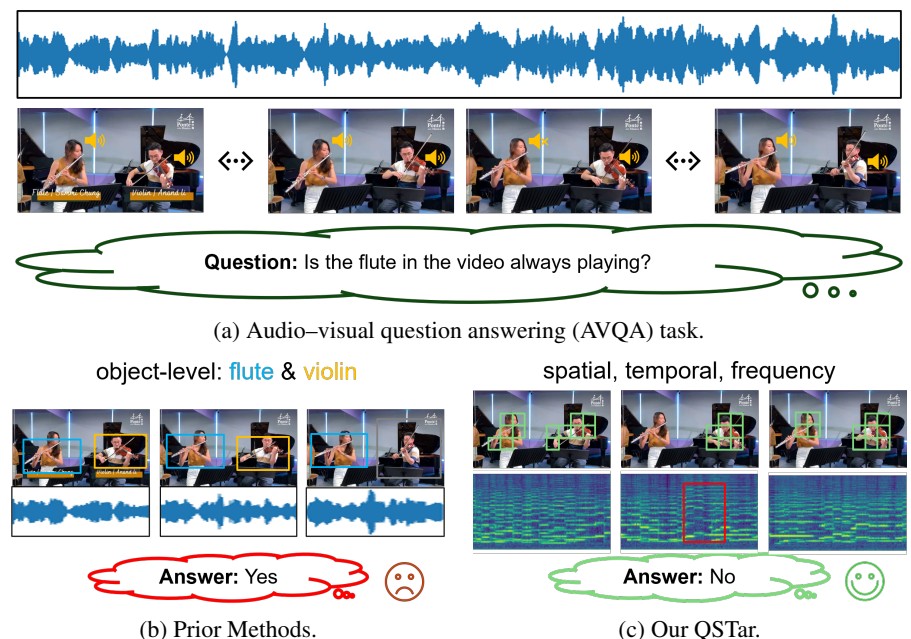

(a) Audio–visual question answering (AVQA) task.

(b) Prior Methods.

(c) Our QSTar.

Figure 1: Illustration of AVQA task and comparison between prior methods and QSTar. (a) Input sample (audio, video, and question) for an AVQA task. (b) Prior works rely on object-level cues, struggling with subtle motions (*e.g.,* inactive flute player). (c) QSTar enhances spatial–temporal–frequency interaction. Green patches highlight spatial focus. The red box shows diminishing high-frequency bands as the flute stops, while the violin remains active.

performances where motion is minimal despite continuous presence), as illustrated in Fig. 1. This emphasizes the value of audio cues in improving question answering accuracy.

Most existing AVQA methods primarily focus on visual information processing. For instance, PSTP (Li et al., 2023a) introduced a spatial–temporal perception module to select *Top-K* frames for alignment. APL (Li et al., 2024b) proposed a question–object and audio–object matching scheme that leverages pre-trained object detectors to enhance visual recognition. However, these approaches primarily rely on visual cues, treating audio as a secondary modality for temporal alignment. For instance, in PSOT (Li et al., 2025), audio was used only to select sound-driven patches that were then aggregated into a largely vision-centric pipeline. As a result, the distinctive properties of the aural modality are underutilized. Furthermore, question information is often incorporated only at the final stage via simple operations (*e.g.,* multiplication), limiting the semantic alignment between the query and the multimodal content. For instance, TSPM (Li et al., 2024a) incorporated the question only at the final fusion and prediction stage, and its auxiliary prompt was used merely to identify key temporal segments. We believe that the late fusion leads to redundant audio–visual representations and hinders model performance for AVQA.

In this paper, we jointly consider query guidance for both audio and visual feature learning and enhance cross-modal interactions across multiple dimensions. This design is motivated by the following observations: 1) questions typically target one or a few instruments, requiring focused rather than holistic audio–visual representation; 2) wind instruments (*e.g.,* clarinet and saxophone) often exhibit subtle visual cues but distinctive spectral characteristics, making frequency-domain analysis more effective, as shown in Fig. 1; 3) instruments of the same category still differ in temporal patterns during performance. Based on these insights, our method emphasizes the role of linguistic cues throughout the pipeline to support fine-grained cross-modal reasoning. We emphasize spatial, temporal, and frequency-domain interactions. These are especially important in polyphonic scenarios where multiple instruments play simultaneously and subtle timbral or harmonic cues (Agostini et al., 2003) cannot be captured by visual or temporal features alone.

To address these limitations, we propose a **Q**uery-guided **S**patial–**T**empor**a**l–**Fr**equency Interaction (**QSTar**) method for the music AVQA task. Specifically, we design a query-guided multimodal correlation module that refines audio and visual features conditioned on the question from the early

stage. This module consists of three components: modality-specific self-enhancing, cross-modal guidance capturing, and information propagation. Next, we introduce a spatial–temporal–frequency interaction module that further enriches the query-guided features by emphasizing semantic relevance across spatial, temporal, and frequency dimensions, particularly beneficial for distinguishing sound patterns over time and frequency. Finally, a query context reasoning block is employed to refine and fuse features before prediction. It uses prompting to incorporate linguistic context and task-specific constraints for more precise reasoning. In summary, our method improves query-guided alignment throughout the entire pipeline and effectively integrates audio and visual cues across spatial, temporal, and frequency domains. We validate our method on the benchmark MUSIC-AVQA and other AVQA datasets and show that our approach significantly outperforms previous Audio QA, Visual QA, Video QA, and AVQA methods.

Our **main contributions** can be summarized as:

- We propose QSTar, a novel framework that integrates query guidance throughout the entire pipeline to refine modality-specific features. By embedding linguistic context early, QSTar enhances both audio and visual representations in a question-aware manner, enabling more precise cross-modal reasoning.

- We introduce a fine-grained interaction module that emphasizes semantic cues across spatial, temporal, and frequency dimensions. This design enhances discriminative understanding, especially in polyphonic scenarios where subtle audio or visual cues are critical.

- We design a reasoning block that injects task-aware linguistic context through prompting to guide final predictions, improving semantic alignment between the question and multi-modal features.

- Extensive experiments demonstrate that QSTar achieves new state-of-the-art performance on the MUSIC-AVQA benchmark.

## 2 RELATED WORK

### 2.1 AUDIO–VISUAL SCENE UNDERSTANDING

Over the past decade, significant progress has been made in multimodal learning, particularly in tasks involving audio–visual scene understanding (Duan et al., 2023). These tasks focus on the aural and visual modalities, which are fundamental components of human perception, given their critical roles in interpreting daily life with semantic, spatial, and temporal coherence. This field encompasses a variety of challenges, including sound source localization (Grumiaux et al., 2022), motion recognition (Kuehne et al., 2011), video parsing (Zhang et al., 1995), semantic segmentation (Li et al., 2023b), and video description (Xu et al., 2016). Recent advances exploit the complementary nature of different modalities to enhance information alignment, facilitating more robust scene reasoning. While unified multimodal models can capture global spatiotemporal representations, further research is needed to enhance fine-grained, task-specific focuses.

### 2.2 AUDIO–VISUAL QUESTION ANSWERING

Audio–Visual Question Answering (AVQA) leverages multimodal information from video content to answer user-posed questions. Compared to other QA tasks (*e.g.,* audio QA, visual QA, and video QA), AVQA is more challenging as it requires integration of multiple modalities to address linguistic queries effectively. In particular, inadequate aural or visual perception can lead to suboptimal or even incorrect predictions. To advance research in this domain, several AVQA datasets have been introduced, including Pano-AVQA (Yun et al., 2021), MUSIC-AVQA (Li et al., 2022), and AVQA (Yang et al., 2022). Recent approaches such as LAVISH (Lin et al., 2023) aim to enhance audio-visual association and improve model training efficiency on large-scale data. APL (Li et al., 2024b) introduces a question-conditioned clue discovery module that uses visual or audio embeddings as queries and attends to the question at a single stage. TSPM (Li et al., 2024a) introduces an auxiliary prompt used in the sequential temporal perception module to identify key temporal audio and visual cues and incorporates the question only at the final fusion stage. PSOT (Li et al., 2025) emphasizes object-level and motion-level visual changes to boost performance. In contrast to prior approaches that primarily focus on visual processing and incorporate question information only at later stages,

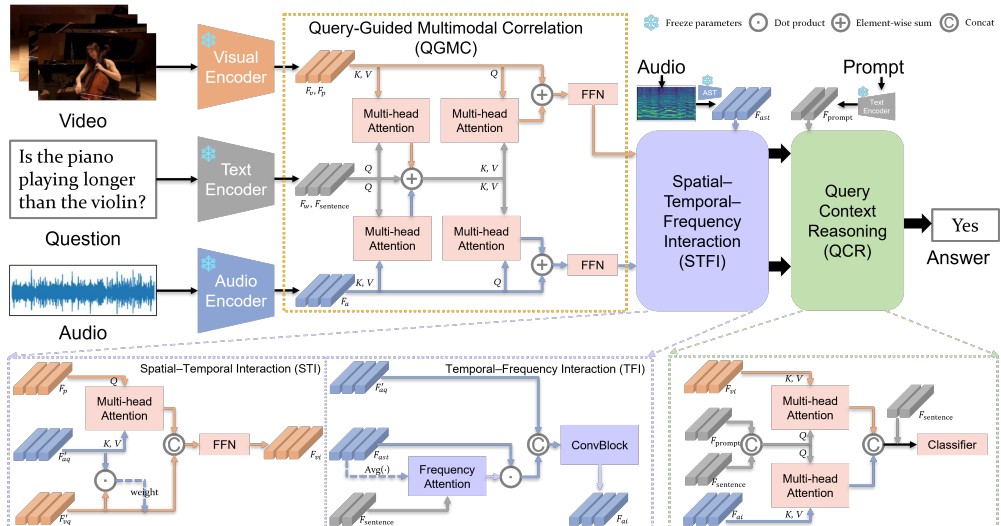

Figure 2: Overall framework of the proposed QSTar method. We use pre-trained encoders to extract audio, visual, and linguistic features, $F_a$, $F_v$, $F_w$, respectively. The Query-Guided Multimodal Correlation module (QGMC in yellow area) refines $F_a$ and $F_v$ using query information, resulting in $F'_{aq}$ and $F'_{vq}$. These features are further enhanced by the Spatial–Temporal–Frequency Interaction module (STFI in purple area), which integrates Spatial–Temporal Interaction (STI) and Temporal–Frequency Interaction (TFI), using additional frequency-aware features from AST (Gong et al., 2021). The Query Context Reasoning block (QCR in green area) incorporates prompt-based context ($F_{\text{prompt}}$) and sentence-level linguistic features ($F_{\text{sentence}}$) to guide multimodal fusion for answer prediction. For brevity, we remove self-attention units.

our method emphasizes richer, query-relevant refinement of both audio and visual features while enabling the model to attend to linguistic cues throughout the entire processing pipeline.

# 3 METHOD

To address the challenges of music AVQA, we propose QSTar, a novel network that enables query-guided extraction of visual and audio cues and facilitates effective spatial–temporal–frequency interaction and query context reasoning, thereby enhancing answer prediction performance. Fig. 2 illustrates the overall framework of the proposed method.

## 3.1 INPUT REPRESENTATION

The input video sequence is first divided into $T$ non-overlapping audio and visual segments, each with a duration of one second. These segments, along with the input question, are then encoded using modality-specific models.

**Visual Representation.** Each visual segment is divided into $M$ patches and appended with a special [CLS] token at the beginning. To effectively represent visual content from the video, we employ a pretrained CLIP (Radford et al., 2021) model with frozen parameters to extract two types of features: frame-level and patch-level representations. The patch level features are further compressed using Token Merging (ToMe) (Bolya et al., 2023), reducing them to $M'$ tokens that preserve spatially sensitive information. Finally, the frame-level and patch-level features across all $T$ video segments are denoted as $F_v = \{f_v^t\}_{t=1}^T \in \mathbb{R}^{T \times D}$ and $F_p = \{f_p^t\}_{t=1}^T \in \mathbb{R}^{T \times M' \times D}$, respectively.

**Audio Representation.** Following previous works, each audio segment is processed using a 2D CNN, VGGish (Gemmeke et al., 2017), which is pretrained on a large-scale AudioSet dataset. The resulting audio features are represented as $F_a = \{f_a^t\}_{t=1}^T \in \mathbb{R}^{T \times D}$.

**Text Representation.** The input question is first tokenized into individual words. We then utilize a pretrained CLIP text encoder to extract both sentence-level and word-level linguistic features. The sentence-level features capture the overall semantic context of the question and are denoted

as $F_{\text{sentence}} \in \mathbb{R}^D$. The word-level features are represented as a fixed-length sequence of token embeddings, denoted as $F_w \in \mathbb{R}^{N \times D}$, where $N$ is the number of tokens.

## 3.2 QUERY-GUIDED MULTIMODAL CORRELATION MODULE

In this subsection, we propose an effective module, **Q**uery-**G**uided **M**ultimodal **C**orrelation (**QGMC**), to enhance unified interaction among visual, audio, and linguistic features. In contrast to the prior two-stage fusion methods that first integrate audio and visual features before incorporating textual information, our approach jointly emphasizes modality-specific features that are semantically aligned with the question. This enables more focused and context-aware multimodal reasoning.

QGMC consists of three stages: *self-enhancing*, *capturing*, and *propagating*. First, internal relationships are strengthened by applying multi-head self-attention (SA) (Vaswani et al., 2017) units independently to the visual, audio, and linguistic features. Next, the self-enhanced word-level linguistic features serve as query in multi-head cross-attention (CA) (Vaswani et al., 2017) units to capture shared semantics from the frame-level visual features and audio features (used as keys and values, respectively). For simplicity, the visual capturing step is formulated as (with a similar operation for audio to obtain $F_{qa}$):

$$F_{qv} = \text{CA}(\text{SA}(F_w), \text{SA}(F_v), \text{SA}(F_v)), \tag{1}$$

To fuse the captured information, we aggregate the cross-modal outputs with residual linguistic features,

$$F_{qg} = F_{qv} + F_{qa} + \text{SA}(F_w), \tag{2}$$

where $F_{qg}$ represents the query-guided semantic context. Subsequently, this guidance is propagated back to the visual and audio streams via cross-attention with $F_v$ as query and $F_{qg}$ as keys and values (or use $F_a$ to obtain $F_{aq}$). This step is denoted as:

$$F_{vq} = \text{CA}(F_v, F_{qg}, F_{qg}), \tag{3}$$

Finally, the outputs are refined by incorporating the original modality features via residual connection, followed by a Feed-Forward Network (FFN):

$$F'_{vq} = \text{FFN}(F_{vq} + F_v); F'_{aq} = \text{FFN}(F_{aq} + F_a), \tag{4}$$

where $F'_{vq}$ and $F'_{aq}$ denote the query-guided visual and audio representations.

## 3.3 SPATIAL–TEMPORAL–FREQUENCY INTERACTION MODULE

To more effectively exploit the audio information in videos, we introduce two dedicated submodules to localize performing instruments across spatial, temporal, and frequency dimensions.

**Spatial–Temporal Interaction.** Since videos contain both spatial and temporal dimensions that may cover redundant information, we design a spatial–temporal submodule to selectively refine the previously obtained query-guided features. Specifically, we enhance the patch-level visual features $F_p$ to better align fine-grained spatial details with the query-guided audio context. First, $F_p$ is refined using a SA unit, and then used as the query in a CA unit, where the query-guided audio features $F'_{aq}$ serve as keys and values:

$$F_{si} = \text{CA}(\text{SA}(F_p), F'_{aq}, F'_{aq}), \tag{5}$$

This operation encourages the model to focus on sounding regions that correspond to the question. In parallel, to capture global temporal dependencies, we perform a temporal interaction between the query-guided audio and visual features. Specifically, we compute the dot product between $F'_{aq}$ and $F'_{vq}$, followed by a softmax function:

$$F_{ti} = F'_{vq} \cdot \text{softmax}(F'^{\top}_{aq} \cdot F'_{vq}), \tag{6}$$

where $\top$ is the transpose operation. We integrate the spatial and temporal outputs by concatenating the corresponding features after reshape, followed by a Feed-Forward Network:

$$F_{vi} = \text{FFN}(\text{Concat}(F_{si}, F_{ti})), \tag{7}$$

where Concat denotes the concatenation layer. This fusion enables the model to yield refined spatial–temporal visual features conditioned on the query.

**Temporal–Frequency Interaction.** In some cases, spatial–temporal perception alone is insufficient for comprehensive music scene understanding. For instance, a flutist typically keeps the instrument near his/her mouth, with minimal or subtle motion that is difficult to detect visually, potentially leading to misinterpretation. In such cases, frequency-based analysis plays a crucial role in distinguishing instruments by capturing their unique spectral signatures. These spectral "fingerprints" are often more informative than time- or spatial-domain features, particularly in polyphonic scenarios or acoustically complex environments for cross-model reasoning.

To better exploit these discriminative clues, we design a temporal–frequency submodule to refine the audio representation. Specifically, we utilize a pretrained Audio Spectrogram Transformer (AST) (Gong et al., 2021) to extract rich frequency-aware features from the audio waveform, denoted as $F_{ast} \in \mathbb{R}^{T \times F \times D}$. Notably, while two instruments may generate similar time-domain waveforms or pitches, their timbral characteristics often differ significantly due to the unique distribution of overtones and harmonics across frequency bands. These subtle spectral distinctions are more effectively captured by AST compared to models like VGGish (Gemmeke et al., 2017). This is largely due to architectural differences: AST's transformers model long-range dependencies across frequency bins, enabling it to represent harmonic structures and overtone patterns that define timbre (Agostini et al., 2003; Zhao et al., 2018; Koutini et al., 2021; Chen et al., 2022). To further highlight informative frequency regions, we introduce a frequency-wise attention over the AST features, enabling the model to explicitly emphasize frequency bands that are most relevant for query-guided instrument recognition. In detail, we first aggregate the AST-based audio features over the temporal dimension to obtain a condensed frequency representation. Using this representation and the question embeddings, we compute frequency attention weights that highlight question-relevant spectral bands. These weights are then broadcast and applied across the original AST features to emphasize salient frequency regions:

$$f_{mean} = \frac{1}{T} \sum_{t=1}^{T} F_{ast}[t, :, :], \tag{8}$$

$$a_f = \mathrm{softmax}(W_1 \cdot F_w + W_2 \cdot \mathrm{ReLU}(W_3 \cdot f_{mean}^{\top})), \tag{9}$$

$$F'_{ast} = a_f \cdot F_{ast}, \tag{10}$$

where $W_1$, $W_2$, and $W_3$ are learnable projection matrices. Finally, to align and integrate $F'_{ast}$ and $F'_{aq}$, we concatenate them and apply a convolutional fusion block composed of two convolutional layers with batch normalization and ReLU activation:

$$F_{ai} = \mathrm{ConvBlock}(\mathrm{Concat}(F'_{ast}, F'_{aq})), \tag{11}$$

This fusion allows the model to effectively associate discriminative spectral patterns with the corresponding instruments, producing more informative and query-guided audio features.

## 3.4 QUERY CONTEXT REASONING BLOCK AND PREDICTION

To enhance final answer prediction using the refined visual and audio features, we introduce a **Q**uery **C**ontext **R**easoning (**QCR**) block inspired by prompt-based conditioning (Khattak et al., 2023). Prompts can provide focused guidance for interpreting visual and audio content under task-specific constraints. In our music scene understanding task, the relevant cues often involve musical instruments, specifically their *type*, *performance duration*, *location*, *temporal sequence*, and *loudness*. These aspects, though derived from the dataset's question types, capture general audio–visual reasoning dimensions. They are used to construct the query context, which guides the refinement of the updated modality-specific features. More details about the prompts can be found in Appendix Sec. A.2.3. We first encode these context-related keywords using the same CLIP text encoder, yielding prompt embeddings denoted as $F_{\mathrm{prompt}}$. Since not all aspects are equally relevant for each question, we incorporate the sentence-level question embedding $F_{\mathrm{sentence}}$. They are concatenated and passed through a multi-head self-attention unit to produce the query context feature $F_{qc}$:

$$F_{qc} = \mathrm{SA}(\mathrm{Concat}(F_{\mathrm{prompt}}, F_{\mathrm{sentence}})), \tag{12}$$

Next, we apply cross-attention using $F_{qc}$ as the query to guide the refinement of visual and audio features $F_{vi}$ and $F_{ai}$, respectively:

$$F_{fv} = \mathrm{CA}(F_{qc}, F_{vi}, F_{vi}); F_{fa} = \mathrm{CA}(F_{qc}, F_{ai}, F_{ai}), \tag{13}$$

This process enables the model to extract the most informative audio and visual cues under the linguistic query context. For final answer prediction, we perform a simple multimodal fusion and classification. The fused representation is obtained by

$$F_{av} = \text{FC}(\text{Concat}(F_{fv}, F_{fa})), \tag{14}$$

where $F_{fv}$ and $F_{fa}$ are combined with a concatenation layer and a linear layer followed by a tanh activation function. Then we adopt an element-wise multiplication operation $\circ$ to integrate the question features and final visual audio representations as: $e = F_{\text{sentence}} \circ F_{av}$. The resulting feature $e$ is then used to predict the final answer from a predefined vocabulary set.

# 4 EXPERIMENTS

## 4.1 DATASETS AND EVALUATION METRIC

**Dataset.** The experiments were primarily conducted on the widely-used MUSIC-AVQA (Li et al., 2022) dataset. MUSIC-AVQA contains more than 40K QA pairs distributed across 9,288 videos. These pairs, centered around music-related scenarios, are categorized into three types based on the modalities required to answer them: Audio QA, Visual QA, and Audio-Visual QA. The questions were generated using a set of predefined templates. We also evaluated models on AVQA (Yang et al., 2022), which comprises over 57K QA pairs derived from real-world videos, and provided discussion on the compared results in Appendix Sec. A.1.

**Evaluation Metric.** Following the standard protocol in previous AVQA studies, we report answer accuracy (Antol et al., 2015) for each question type (as classified in the benchmark) along with the overall average to evaluate model performance in different audio–visual scenarios.

## 4.2 IMPLEMENTATION DETAILS

Videos were segmented into 1-second clips from 60-second recordings. We used a CLIP-ViT-L/14 (Radford et al., 2021) model for visual and textual feature extraction. Audio features were extracted using VGGish (Gemmeke et al., 2017) pre-trained on AudioSet, while AST (Gong et al., 2021) was employed to capture time–frequency representations. All features were projected into 512-dimensional vectors for consistency. We trained all models using the AdamW (Loshchilov et al., 2017) optimizer with an initial learning rate of 1$e$-4, decayed by a factor of 0.1 every 10 epochs. The batch size and number of training epochs were set to 64 and 30, respectively. All experiments were conducted using PyTorch on a single NVIDIA H100 GPU.

## 4.3 COMPARISON WITH STATE-OF-THE-ART METHODS

**Compared Methods.** To verify the effectiveness of the proposed method, we compared it with previous state-of-the-art multimodal QA approaches. For **audio QA**, we adopted FCN-LSTM (Fayek & Johnson, 2020). For **visual QA**, we considered MCAN (Yu et al., 2019). For **video QA**, we evaluated PSAC (Li et al., 2019), HME (Fan et al., 2019), and HCRN (Le et al., 2020). Finally, we compared against a comprehensive set of recent **AVQA** methods, including AVSD (Schwartz et al., 2019), PanaAVQA (Yun et al., 2021), AVST (Li et al., 2022), COCA (Lao et al., 2023), PSTP (Li et al., 2023a), LAVISH (Lin et al., 2023), APL (Li et al., 2024b), TSPM (Li et al., 2024a), MCCD (Ma et al., 2024), PSOT (Li et al., 2025), and QA-TIGER (Kim et al., 2025). All methods were trained and evaluated using the same dataset split for fair comparison. We further compared our method with recent large multimodal models (MLLMs), including GPT-4o (Hurst et al., 2024), VideoLLaMA2 (Cheng et al., 2024), Qwen2.5-Omni (Xu et al., 2025), and Ming-Omni (AI et al., 2025).

**Quantitative Results.** The quantitative results are reported in Table 1. Our proposed QSTar achieved state-of-the-art results across most question types, outperforming TSPM and QA-TIGER by 2.19% and 1.36% in overall accuracy, respectively. Compared with earlier models designed for single-modality reasoning with simple multimodal fusion (*e.g.,* MCAN for visual QA and HCRN for video QA), QSTar showed substantial gains across all metrics, which are consistent with trends observed in other AVQA models. These improvements underscore the advantage of deep multimodal alignment among audio, visual, and textual features for complex video scene understanding.

Table 1: Comparison with existing methods on the MUSIC-AVQA (Li et al., 2022) test set, reporting accuracy (%) across different question types. The short names are the abbreviations for question types "Counting", "Comparative", "Location", "Existential", and "Temporal", respectively. For space, FCN-LSTM, VideoLLaMA2, and Qwen2.5-Omni are denoted as FCN, VLLaMa2, and Qwen-Omni, respectively. The best results are **bold** while the second best are underlined.

| Method | Audio QA | | | Visual QA | | | Audio-Visual QA | | | | | | Avg |
| --- | --- | --- | --- | --- | --- | --- | --- | --- | --- | --- | --- | --- | --- |
| | Count. | Comp. | Avg | Count. | Local. | Avg | Exist. | Count. | Local. | Comp. | Temp. | Avg | |
| FCN (Fayek & Johnson, 2020) | 70.80 | 65.66 | 68.90 | 64.58 | 48.08 | 56.23 | 82.29 | 59.92 | 46.20 | 62.94 | 47.45 | 60.42 | 60.81 |
| MCAN (Yu et al., 2019) | 78.07 | 57.74 | 70.58 | 71.76 | 71.76 | 71.76 | 80.77 | 65.22 | 54.57 | 56.77 | 46.84 | 61.52 | 65.83 |
| PSAC (Li et al., 2019) | 75.02 | 66.84 | 72.00 | 68.00 | 70.78 | 69.41 | 79.76 | 61.66 | 55.22 | 61.13 | 59.85 | 63.60 | 66.62 |
| HME (Fan et al., 2019) | 73.65 | 63.74 | 69.89 | 67.42 | 70.20 | 68.83 | 80.87 | 63.64 | 54.89 | 63.03 | 60.58 | 64.78 | 66.75 |
| AVSD (Schwartz et al., 2019) | 72.47 | 62.46 | 68.78 | 66.00 | 74.53 | 70.31 | 80.77 | 64.03 | 57.93 | 62.85 | 61.07 | 65.44 | 67.32 |
| HCRN (Le et al., 2020) | 71.29 | 50.67 | 63.69 | 65.33 | 64.98 | 65.15 | 54.15 | 53.28 | 41.74 | 51.04 | 46.72 | 49.82 | 56.34 |
| LAViT (Yun et al., 2021) | 75.71 | 65.99 | 72.13 | 70.51 | 75.76 | 73.16 | 82.09 | 65.38 | 61.30 | 63.67 | 62.04 | 66.97 | 69.53 |
| AVST (Li et al., 2022) | 77.78 | 67.17 | 73.87 | 73.52 | 75.27 | 74.40 | 82.49 | 69.88 | 64.24 | 64.67 | 65.82 | 69.53 | 71.59 |
| COCA (Lao et al., 2023) | 79.35 | 67.68 | 75.42 | 75.10 | 75.43 | 75.23 | 83.50 | 66.63 | 69.72 | 64.12 | 65.57 | 69.96 | 72.33 |
| PSTP (Li et al., 2023a) | 73.97 | 65.59 | 70.91 | 77.15 | 77.36 | 77.26 | 76.18 | 72.23 | 71.80 | **71.79** | 69.00 | 72.57 | 73.52 |
| LAVISH (Lin et al., 2023) | 82.09 | 65.56 | 75.97 | 78.98 | 81.43 | 80.22 | 81.71 | 75.51 | 66.13 | 63.77 | 67.96 | 71.26 | 74.46 |
| APL (Li et al., 2024b) | 82.40 | 70.71 | 78.09 | 76.52 | 82.74 | 79.69 | 82.99 | 73.29 | 66.68 | 64.76 | 65.95 | 70.96 | 74.53 |
| VLLaMa2 (Cheng et al., 2024) | 79.65 | 52.69 | 69.71 | 81.20 | 83.02 | 82.12 | 77.43 | 63.48 | 69.67 | 62.67 | 68.13 | 67.88 | 71.98 |
| GPT-4o (Hurst et al., 2024) | 65.68 | 37.04 | 55.12 | 72.77 | 62.20 | 67.42 | 55.87 | 54.94 | 59.57 | 38.24 | 42.58 | 50.35 | 55.72 |
| TSPM (Li et al., 2024a) | 84.07 | 64.65 | 76.91 | 82.29 | 84.90 | 83.61 | 82.19 | 76.21 | 71.85 | 65.76 | 71.17 | 73.51 | 76.79 |
| MCCD (Ma et al., 2024) | 83.87 | 71.04 | 79.14 | 79.78 | 76.73 | 78.24 | 80.87 | 71.46 | 51.63 | 64.67 | 64.60 | 67.13 | 72.20 |
| PSOT (Li et al., 2025) | - | - | 78.22 | - | - | 80.07 | - | - | - | - | - | 72.61 | 75.29 |
| Qwen-Omni (Xu et al., 2025) | 62.93 | 39.56 | 54.31 | 67.92 | 59.02 | 63.42 | 51.92 | 55.10 | 60.98 | 36.42 | 40.27 | 49.12 | 53.83 |
| Ming-Omni (AI et al., 2025) | 62.44 | 40.74 | 54.44 | 67.50 | 58.29 | 62.84 | 52.73 | 53.91 | 62.17 | 37.24 | 41.85 | 49.63 | 53.98 |
| QA-TIGER (Kim et al., 2025) | 84.86 | 67.85 | 78.58 | **83.96** | **86.29** | **85.14** | 83.10 | 78.58 | 72.50 | 63.94 | 69.59 | 73.74 | 77.62 |
| QSTar (ours) | **85.64** | **72.05** | **80.63** | 83.46 | 84.90 | 84.17 | **83.81** | **79.76** | **72.72** | 70.03 | **72.38** | **75.98** | **78.98** |

Notably, QSTar surpassed existing AVQA approaches that primarily rely on visual processing techniques such as frame selection or object-level perception. For example, QSTar achieved 78.98% average accuracy, compared to 76.79% by TSPM and 74.53% by APL. It also outperformed the previous SOTA QA-TIGER by 2.05% on Audio QA and 2.24% on Audio-Visual QA types. For example, in Audio QA, our method achieved a 4.2% gain over QA-TIGER on comparative questions. In Audio-Visual QA, comparative and temporal questions improved by 6.09% and 2.79%, respectively, compared with QA-TIGER. These gains arise because frequency energy patterns provide precise indicators of instrument activity, especially for polyphonic scenes. Unlike visual cues, which may be subtle, occluded, or ambiguous, frequency features clearly reflect when an instrument starts, stops, or changes intensity. These results demonstrate the strength of our frequency-aware modeling design. Despite not relying on pre-trained object detectors or specially designed visual perception modules, QSTar still performed competitively on Visual QA type, trailing QA-TIGER by only 0.97%, while significantly outperforming other vision-centric methods such as APL (+4.48%) and PSOT (+4.1%). This demonstrates that our query-guided multimodal design preserves strong visual reasoning. The comparison highlights the effectiveness of our model and motivates future enhancements in the spatial localization of performing instruments.

In addition, we conducted experiments and report comparisons with GPT-4o, Qwen2.5-Omni, and Ming-Omni. Following common practice for proprietary models, these MLLMs were evaluated in a zero-shot setting due to their closed-source nature and large-scale pre-training benefits. The highly capable GPT-4o achieved only 55.72% average accuracy while Qwen2.5-Omni (7B) and Ming-Lite-Omni achieved similar modest results at 53.83% and 53.98%, respectively. These models struggled particularly with question types requiring fine-grained temporal and comparative reasoning in polyphonic scenes, exhibiting drops of nearly 30% on these categories. In contrast, QSTar significantly outperformed all evaluated zero-shot MLLMs across every question type. We further assessed MLLM performance under a fine-tuning setting by comparing against VideoLLaMA2, a representative open-source MLLM. Even with task-specific fine-tuning, VideoLLaMA2 still trailed QSTar substantially—particularly on comparative questions, where QSTar exceeded it by 19.36% (Audio QA) and 7.36% (Audio-Visual QA). This performance gain highlights the specialized strength of QSTar on cross-modal reasoning within complex polyphonic scenes.

Table 2: Ablation study on the proposed main modules.

| Method | Audio QA | Visual QA | A-V QA | Avg |
|---|---|---|---|---|
| *w/o* all | 73.87 | 79.15 | 70.33 | 73.29 |
| *w/o* QGMC | 79.08 | 83.44 | 72.92 | 76.80 |
| *w/o* QCR | 79.33 | 83.24 | 75.43 | 78.19 |
| *w/o* STI | 79.21 | 82.62 | 75.06 | 77.80 |
| *w/o* TFI | 78.21 | 83.24 | 74.39 | 77.41 |
| *w/o* STFI | 77.09 | 81.79 | 74.02 | 76.62 |
| QSTar | **80.63** | **84.17** | **75.98** | **78.98** |

Table 3: Ablation study on the query guidance. The notations *r/m* in B, M, and F represent the exclusion of query guidance during the beginning, middle, and final stages, respectively.

| Method | Audio QA | Visual QA | A-V QA | Avg |
|---|---|---|---|---|
| *r/m* in B | 78.96 | 83.61 | 74.90 | 77.93 |
| *r/m* in M | 80.07 | 83.65 | 75.65 | 78.55 |
| *r/m* in F | 79.39 | 83.32 | 75.47 | 78.25 |
| QSTar | **80.63** | **84.17** | **75.98** | **78.98** |

Table 4: Efficiency analysis and comparison on MUSIC-AVQA test set.

| Method | Trainable Parameters (M) | FLOPs (G) | Avg Accuracy (%) |
|---|---|---|---|
| AVST (Li et al., 2022) | 18.48 | 3.19 | 71.59 |
| LAVISH (Lin et al., 2023) | 21.09 | - | 74.46 |
| TSPM (Li et al., 2024a) | 6.22 | 1.42 | 76.79 |
| QA-TIGER (Kim et al., 2025) | 14.51 | 2.70 | 77.62 |
| QSTar (*w/o* TFI) | 11.95 | 2.15 | 77.41 |
| QSTar (ours) | 13.20 | 2.43 | **78.98** |

## 4.4 ABLATION STUDIES

**Ablation Study on Main Modules.** To verify the effectiveness of the main modules (*i.e.,* QGMC, STI, TFI, QCR, *etc.*), we conducted ablation studies by removing each module. Table 2 reports the corresponding results. Removing all modules and retaining only simple multimodal fusion led to a performance drop of over 5%, highlighting the overall importance of our design. Specifically, removing QGMC and QCR reduced the overall accuracy to 76.80% and 78.19%, respectively. Performance dropped across all question types as well, underscoring the importance of multimodal feature correlation and context-aware reasoning tailored to each query. The removal of the spatial–temporal–frequency interaction module also significantly impacted performance. Without STI, overall accuracy dropped by 1.18%, and notably, Visual QA accuracy declined by 1.55%, emphasizing the value of our spatial perception and temporal alignment. Eliminating the TFI module caused a sharp decrease in Audio QA and Audio-Visual QA performance (2.42% and 1.59%, respectively), indicating that questions requiring recognition of which instrument is sounding or when it becomes active rely heavily on frequency-domain cues. The improvement is particularly notable for comparative and temporal questions, where reasoning depends on detecting onset/offset transitions that are often imperceptible in RGB frames. These results demonstrate the necessity of frequency-domain reasoning for auditory understanding. In summary, each module plays a critical role in enhancing model performance. When integrated, they work synergistically, enabling QSTar to achieve the best results on the MUSIC-AVQA benchmark. More ablation studies on these modules and their alternatives can be found in Appendix Sec. A.2.

**Effect of Query Guidance throughout the Pipeline.** To assess the impact of query guidance throughout the QSTar pipeline, we performed ablation studies by selectively removing its corresponding components. Query guidance is integrated at three stages: beginning (via query-guided semantic context in QGMC), middle (through question embeddings in TFI), and final (using prompting in QCR). The resulting ablated models are denoted as $M_b^-$, $M_m^-$, and $M_f^-$, respectively. Note that the final-stage multiplication, as commonly used in prior works, was retained in all ablations. The results, shown in Table 3, indicate consistent performance drops when any stage of query guidance is removed. Removing early-stage guidance ($M_b^-$) resulted in a 1.05% drop, indicating that early query-guided refinement helps identify representative audio and visual features. In the middle stage, $M_m^-$, which omits question embeddings in TFI, led to only a slight drop (0.43%), suggesting that prior query-guided features already carried useful linguistic information. Final-stage prompting also proved beneficial, as $M_f^-$ underperformed QSTar (by 0.73%), confirming the value of incorporating query context during reasoning. These findings underscore the importance of integrating query guidance throughout the pipeline to enhance audio–visual scene understanding. Additional studies on the prompting mechanism are provided in Appendix Sec. A.2.3.

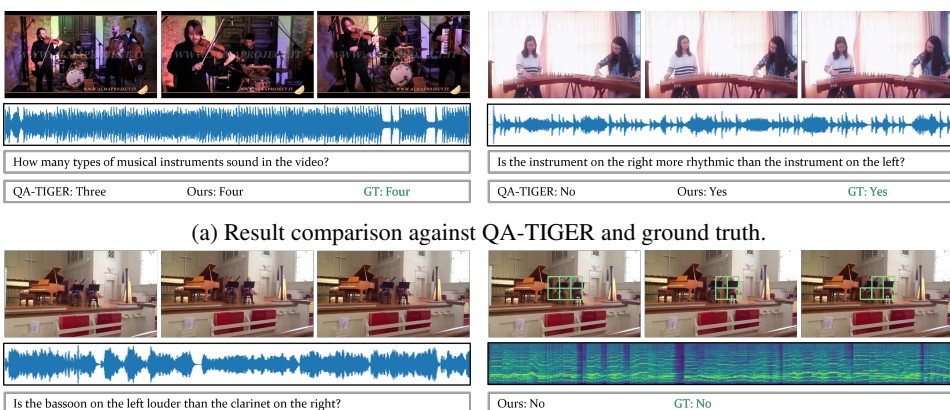

(a) Result comparison against QA-TIGER and ground truth.

(b) One example of QSTar prediction (Left: input sample; Right: prediction and question-related sounding area and timestamps).

Figure 3: Qualitative results in the MUSIC-AVQA (Li et al., 2022) dataset. (a) compares answer predictions between QSTar and QA-TIGER (Kim et al., 2025) on two examples. (b) visualizes spatial, temporal, and frequency focuses from QSTar's STFI module. Green boxes highlight the patch-level visual attention at key timestamps selected via audio-based temporal focus. A 2D spectrogram provides an overview of frequency dynamics for better interpretability.

## 4.5 COMPUTATIONAL COSTS

Table 4 reports the computational costs of QSTar compared with AVST, LAVISH, TSPM, and QA-TIGER. QSTar used fewer trainable parameters, lower FLOPs, and achieved higher accuracy than AVST and LAVISH. Compared to QA-TIGER, QSTar delivered higher performance while maintaining comparable computational costs. Although TSPM is the most lightweight, its accuracy lagged behind QSTar by 2.19%, indicating that QSTar provides a substantially better accuracy–efficiency trade-off. Despite introducing an additional AST-based audio branch, QSTar remains efficient because it employs a lightweight frequency-interaction module (TFI). As shown in the table, this led to an increase of 1.25M trainable parameters and 0.28G FLOPs compared to QSTar without TFI, while still keeping the overall complexity comparable to or lower than other models. The AST branch introduced only a modest computation overhead for a clear accuracy gain (1.57%).

## 4.6 QUALITATIVE RESULTS

In Fig. 3, we present qualitative results from the MUSIC-AVQA test set. Subfigure (a) demonstrates QSTar's advantage over the previous SOTA method, QA-TIGER, in complex multi-instrument scenarios. For instance, even when the cello is not consistently visible, QSTar correctly predicted the answer *Four* by leveraging frequency-enhanced audio cues. Similarly, our method succeeded in distinguishing two guzhengs in a performance scene. Subfigure (b) visualizes temporal visual attentions and frequency-relevant audio information. The STFI module captures dynamic changes in audio intensity, correctly identifying that the clarinet continues playing while the bassoon stops at the middle timestamp. These examples highlight QSTar's ability to localize query-relevant instruments across space, time, and frequency, enabling accurate reasoning in complex musical scenes.

## 5 CONCLUSION

In this work, we propose a **Q**uery-Guided **S**patial–**T**empor**a**l–**F**requency Interaction (QSTar) method for the music AVQA task. We design a query-guided multimodal correlation module that incorporates the query guidance into audio–visual feature learning to enhance relevant representations. We also propose a spatial, temporal, and frequency interaction module to effectively align audio, visual, and textual modalities across these dimensions. Furthermore, a prompting-based query reasoning block is employed to incorporate linguistic context before final prediction. Extensive experiments on the standard AVQA datasets demonstrate that QSTar consistently outperforms state-of-the-art methods, while ablation studies confirm the effectiveness of its components.

ACKNOWLEDGEMENT

This work is partially funded by the European Union within the Horizon Europe research and innovation programme under grant agreement No. 101136006 – XTREME project, coordinated by S.S. Brandt/IT University of Copenhagen, Denmark. Views and opinions expressed are however those of the author(s) only and do not necessarily reflect those of the European Union. European Union can not be held responsible for them. The work is also supported in part by the Pioneer Centre for AI (DNRF grant number P1).

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

# A APPENDIX

This supplementary document includes detailed discussion of the results on AVQA (Yang et al., 2022) dataset, additional ablation studies on main modules, and more qualitative results across different question types. We also provide a discussion for future work.

## A.1 RESULTS ON AVQA DATASET

To further assess the robustness of our proposed method, we evaluated it on the AVQA dataset, which includes more diverse scenarios beyond music-related events. In detail, AVQA covers diverse everyday scenes (e.g., alarms, dog barking, speech, water flow, and vehicle sounds). These signals are typically broadband, less structured, and often non-harmonic. The corresponding questions generally focus on salient, dominant audio events (e.g., "Is the alarm ringing?") rather than subtle harmonic differences, compared to MUSIC-AVQA. To ensure direct comparability and mitigate potential implementation discrepancies, we thus follow the exact reporting format established by prior methods (Li et al., 2023a; 2024a) in Table 5. We also report our model's performance with and without query prompting for fair comparison. Our method outperformed previous PSTP and TSPM in terms of average accuracy, demonstrating that the proposed modules are not restricted to music-specific settings.

## A.2 MORE ABLATION STUDIES

In this section, we further report several groups of ablation studies with respect to the designs of query-guided multimodal correlation (QGMC) module, spatial–temporal–frequency interaction (STFI) module, and query context reasoning (QCR) block.

### A.2.1 ABLATION ON QUESTION-GUIDED FEATURE PROCESSING

To evaluate the impact of different strategies for early-stage multimodal feature processing, we conducted an ablation study focused on question-guided representations. We compared three alternatives: (a) direct fusion of audio and visual features, as in AVST (Li et al., 2022); (b) separate cross-attention between linguistic features and each modality (audio/visual); and (c) sequential cross-attention with audio and linguistic cues for visual update (similar for audio), as in QA-TIGER (Kim et al., 2025). Table 6 presents the results. All methods incorporating early question guidance outperformed (a), with at least a 0.8% gain in average accuracy. Method (b) confirms the utility of basic cross-attention for enhancing modality-specific representations. While (c) achieved a slight edge over our method (d) on Audio-Visual QA type (0.08%), it did not surpass our approach in overall performance. These findings highlight the effectiveness of question-guided processing and suggest that early fusion of audio and visual features offers limited additional benefit.

### A.2.2 ABLATION ON FEATURE USAGE IN STFI

To assess the contribution of the additional features introduced beyond the query-guided representations from QGMC, we conducted an ablation study by removing patch-level visual features ($F_p$) and AST-based (Gong et al., 2021) audio features ($F_{ast}$) from the spatial—temporal interaction (STI) and temporal—frequency interaction (TFI) modules, respectively. The results are reported in Table 7. Excluding $F_p$ led to a 1.43% and 0.9% drop in Visual QA and Audio-Visual QA performance, confirming the finding that patch-level features are critical for spatial perception. Removing $F_{ast}$ caused a 1.37% decrease in average accuracy, notably degrading performance on audio-related questions. This aligns with recent work (Pei et al., 2025) showing that AST provides richer audio representations than VGGish (Gemmeke et al., 2017), especially in complex audio–video scenes. We attribute the improved performance to our frequency-wise audio modeling, which enhances the model's ability to distinguish instruments both within and across categories.

### A.2.3 ABLATION ON PROMPTING IN QCR

Before introducing the prompts used in our method, we first examine the MUSIC-AVQA (Li et al., 2022) benchmark to motivate the design of our query context. Specifically, we identify five key aspects (type, duration, location, sequence, and loudness) as fundamental to understanding music

Table 5: Comparison with existing methods on the AVQA (Yang et al., 2022) test set.

| Method | Prompting | Total Accuracy (%) |
|---|---|---|
| PSAC (Li et al., 2019) | | 87.4 |
| HCRN (Le et al., 2020) | | 89.0 |
| PSTP (Li et al., 2023a) | | 90.2 |
| TSPM (Li et al., 2024a) | ✓ | 90.8 |
| QSTar (ours) | | 90.9 |
| QSTar (ours) | ✓ | **91.2** |

Table 6: Ablation study on early-stage question-guided feature processing. (a) integrates audio and visual features to form sounding visual features (AVST (Li et al., 2022)); (b) applies separate cross-attention between linguistic and each modality; (c) uses dual-branch sequential cross-attention (QA-TIGER (Kim et al., 2025)); (d) our proposed query-guided multimodal correlation (QGMC) module.

| Method | Audio QA | Visual QA | Audio-Visual QA | Avg |
|---|---|---|---|---|
| a | 78.03 | 83.69 | 74.43 | 77.52 |
| b | 80.20 | 83.86 | 75.06 | 78.30 |
| c | 80.07 | 84.06 | **76.06** | 78.89 |
| d (ours) | **80.63** | **84.17** | 75.98 | **78.98** |

scenes and guiding multimodal reasoning. The MUSIC-AVQA dataset features questions involving 22 instruments, categorized into four types: String (*i.e., violin, cello, guitar, ukulele, erhu, guzheng, pipa, bass, banjo*), Wind (*i.e., tuba, trumpet, suona, bassoon, clarinet, bagpipe, flute, saxophone*), Percussion (*i.e., drum, xylophone, congas*), and Keyboard (*i.e., accordion, piano*). Each instrument class involves distinct visual and auditory cues, and depending on the user's question, different modalities and temporal spans may become more relevant for accurate reasoning. We present all 33 sample questions from MUSIC-AVQA in Table 8, along with a summary of the key analytical aspects they target. Most questions focus on one to three core elements of music scene understanding. Accordingly, our prompt-based query context construction uses a set of guiding keywords (*type*, *performance duration*, *location*, *temporal sequence*, and *loudness*) to capture these aspects effectively. These prompts explicitly correspond to these reasoning dimensions: identifying what is sounding or appearing, when an event occurs, where it occurs, in what order, and with what intensity. For example, under the "type" aspect, the underlying focus may concern instrument category or audio identity. Note that we do not craft question-specific prompts or explanations. We intentionally use a unified prompt formulation for two reasons: (1) Avoiding prompt-answer leakage. Using tailored prompts per question risks encoding partial answers or shortcuts. Unified prompts ensure the model must rely on audio, visual, and textual cues during reasoning. (2) Ensuring scalability and stability. A consistent, domain-agnostic prompt set scales naturally to new question types and datasets, while also providing stable semantic anchors for multimodal alignment.

To evaluate the effectiveness of our proposed prompts, we conducted an ablation study on MUSIC-AVQA. As a baseline, we removed the prompts entirely, following the $M_f^-$ variant described in the main text. We then compared three alternative prompting strategies: (1) the declarative prompt formulation used in TSPM (Li et al., 2024a), which converts questions into related statements. For instance, a given question "Where is the first sounding instrument?" is converted to "The instruments in the video do not sound at the same time."; (2) prompts generated using a pre-trained video captioning model, Scenic (Zhou et al., 2024), to describe overall video dynamics as a longer narrative

Table 7: Ablation study on the feature usage in the spatial–temporal–frequency interaction module.

| Method | Audio QA | Visual QA | Audio-Visual QA | Avg |
|---|---|---|---|---|
| *w/o $F_p$* | 79.27 | 82.74 | 75.08 | 77.88 |
| *w/o $F_{ast}$* | 78.90 | 83.24 | 74.53 | 77.61 |
| STFI (ours) | **80.63** | **84.17** | **75.98** | **78.98** |

Table 8: Overview of the five question types and 33 sample questions from MUSIC-AVQA (Li et al., 2022), along with their summarized focus areas (*i.e.,* type, duration, sequence, location, and loudness) for the answering task.

| Question Types | Sample Questions | Focuses |
|---|---|---|
| Counting | Is there a clarinet sound? | type & loudness |
| | How many musical instruments were heard throughout the video? | type & loudness |
| | How many types of musical instruments were heard throughout the video? | type & loudness |
| | Is there a violin in the entire video? | type |
| | Are there drum and piano instruments in the video? | type |
| | How many types of musical instruments appeared in the entire video? | type |
| | How many guzhengs are in the entire video? | type |
| | How many instruments are sounding in the video? | type & loudness |
| | How many types of musical instruments sound in the video? | type & loudness |
| | How many instruments in the video did not sound from beginning to end? | type & duration |
| | How many sounding sounas in the video? | type & loudness |
| Comparative | Is the guitar more rhythmic than the cello? | type & sequence |
| | Is the clarinet louder than the bassoon? | type & loudness |
| | Is the saxophone playing longer than the banjo? | type & duration |
| | Is the instrument on the left more rhythmic than the instrument on the right? | location & sequence |
| | Is the instrument on the right louder than the instrument on the left? | location & loudness |
| | Is the pipa on the left more rhythmic than the erhu on the right? | type & location & sequence |
| | Is the ukulele on the right louder than the bass on the left? | type & location & loudness |
| Location | Where is the performance? | location |
| | What is the instrument on the left of xylophone? | type & location |
| | What kind of musical instrument is it? | type |
| | What kind of instrument is the leftmost instrument? | type & location |
| | Where is the loudest instrument? | location & loudness |
| | Is the first sound coming from the right instrument? | location & sequence |
| | Which is the musical instrument that sounds at the same time as the congas? | type & sequence |
| | What is the right instrument of the last sounding instrument? | type & location & sequence |
| Existential | Is this sound from the instrument in the video? | loudness |
| | Is the accordion in the video always playing? | type & duration |
| | Is there a voiceover? | type & loudness |
| Temporal | Where is the first sounding instrument? | location & sequence |
| | Which pipa makes the sound last? | type & sequence |
| | Which instrument makes sounds before the trumpet? | type & sequence |

Table 9: Ablation study on alternative prompting strategies for query-context construction. Translation refers to question-to-statement conversions used in TSPM (Li et al., 2024a); Caption denotes video-level descriptions generated by a pretrained captioning model (Zhou et al., 2024); Generative Prompts correspond to constrained attribute-oriented texts produced by GPT-4o (Hurst et al., 2024).

| Method | Audio QA | Visual QA | Audio-Visual QA | Avg |
|---|---|---|---|---|
| *w/o* Prompts | 79.39 | 83.32 | 75.47 | 78.25 |
| *w* Translation | 79.45 | 83.77 | 75.59 | 78.44 |
| *w* Caption | 79.33 | 84.35 | 75.06 | 78.28 |
| *w* Generative Prompts | 79.21 | **84.56** | 75.16 | 78.37 |
| QCR (ours) | **80.63** | 84.17 | **75.98** | **78.98** |

paragraph (*e.g.,* "A man is strumming a guitar while a girl plays the bass beside him in a room."); (3) attribute-expansion prompts, commonly adopted in text-to-image generation. We use the following generative-style prompt template: "A detailed description of instruments, their location, appearance, and actions that help answer the <question>." The produced long-form natural-language descriptions by GPT-4o (Hurst et al., 2024) are encoded using the same pre-trained text encoder as in the other baselines. Table 9 presents the results. Models using prompting strategies consistently outperformed the baseline. Variant (1), which uses temporal translations, underperformed due to its limited focus on temporal cues, failing to capture location-based or comparative queries. Variant (2), aided by global visual summaries, performed better on Visual QA type but struggled with audio-related questions. Variant (3), although attribute-expanded prompts provide more constrained descriptions than free-form captions, they still introduce substantial irrelevant details and dilute query-specific information. Consequently, they did not outperform our unified prompts. Even worse, they may risk prompt-answer leakage. These findings highlight the effectiveness of our tailored prompting approach in capturing diverse query contexts.

Table 10: Ablation study on the temporal audio pathway.

| Method | Audio QA | Visual QA | Audio-Visual QA | Avg |
|---|---|---|---|---|
| w $F_{ast}$ | 78.96 | 83.94 | 75.12 | 78.14 |
| w $F_a$ (ours) | **80.63** | 84.17 | **75.98** | **78.98** |
| w $F_{ast}$ & $F_a$ | 79.83 | **84.19** | 75.65 | 78.65 |

### A.2.4 ABLATION ON FEATURE USAGE FOR AUDIO PATHWAY

In QSTar, the temporal audio pathway is designed to capture coarse temporal dynamics for input audios. For this purpose, VGGish-style convolutional features are sufficient and, importantly, provide stable frame-level embeddings aligned to video timestamps. In contrast, AST provides fine-grained spectral modeling with long-range attention over frequency bins, which is essential for distinguishing harmonic structures, overtone distributions, and instrument timbre. This is why AST features are specifically used inside the TFI module, where frequency reasoning is required. To directly assess this, we include an additional ablation where we replaced the temporal VGGish pathway with AST-only features. As shown in Table 10, this substitution did not improve performance and in fact slightly reduced accuracy due to poorer temporal alignment and feature redundancy. We extended this ablation by adding a variant that combined VGGish and AST features as the feature extraction. The results show that simply merging the two feature sets did not lead to performance improvement. The overall accuracy (78.65%) remained very close to the our version (78.98%), and the Audio QA accuracy even dropped slightly (-0.8%). The observations further support our design choice of assigning distinct roles to the two backbones: VGGish for temporal audio pathway and AST for fine-grained frequency reasoning.

### A.3 QUALITATIVE RESULTS

In this section, we present the predictive results of our method on each question type of the MUSIC-AVQA (Li et al., 2022) dataset in Fig. 4 and Fig. 5. For each example, we show video frames, audio waveforms, the question, our model's top-3 predictions, and the ground truth. These results highlight our model's ability to interpret diverse audio–visual cues and generate accurate answers. Notably, even in challenging cases involving rare instruments (*e.g.,* guzheng, erhu), our method demonstrates strong reasoning and robust performance. Overall, these examples showcase the model's generalization ability across question types and confirm its effectiveness in music scene understanding.

### A.4 DISCUSSION AND FUTURE WORK

Although our proposed method has achieved strong performance in music-scene AVQA, it still has some directions that can be explored in the future. The current model operates on fixed-length video segments and may struggle with longer videos that require temporal abstraction or memory mechanisms. In this work, our frequency-domain modeling is applied only to the audio modality, as audio frequency patterns (timbre, harmonics, onset energy) provide discriminative cues that are unavailable in visual frames. In turn, explicit visual frequency-domain processing (e.g., high-pass/low-pass filtering or DCT-based feature extraction) may help capture micro-motions that are hard to detect in the RGB and audio domains, which is an interesting future direction. Another promising direction is integrating dynamic prompting or large language model-based reasoning, which could enable more flexible and interpretable context modeling. We plan to extend the model to handle open-ended questions beyond the current predefined setup, allowing for richer natural language understanding. For Visual QA, especially location-related questions, our proposed method still slightly lags behind the state-of-the-art. We believe there is substantial room for improvement by integrating object-level cues (e.g., object detectors, motion-based visual features, or segmentation cues) and richer relational reasoning mechanisms such as scene graphs, which could strengthen fine-grained spatial understanding. Last but not least, expanding the approach to other multimodal QA benchmarks (*e.g.,* movie scenes and open-world scenes with complex sound compositions) beyond music-related datasets will further validate its generalizability.

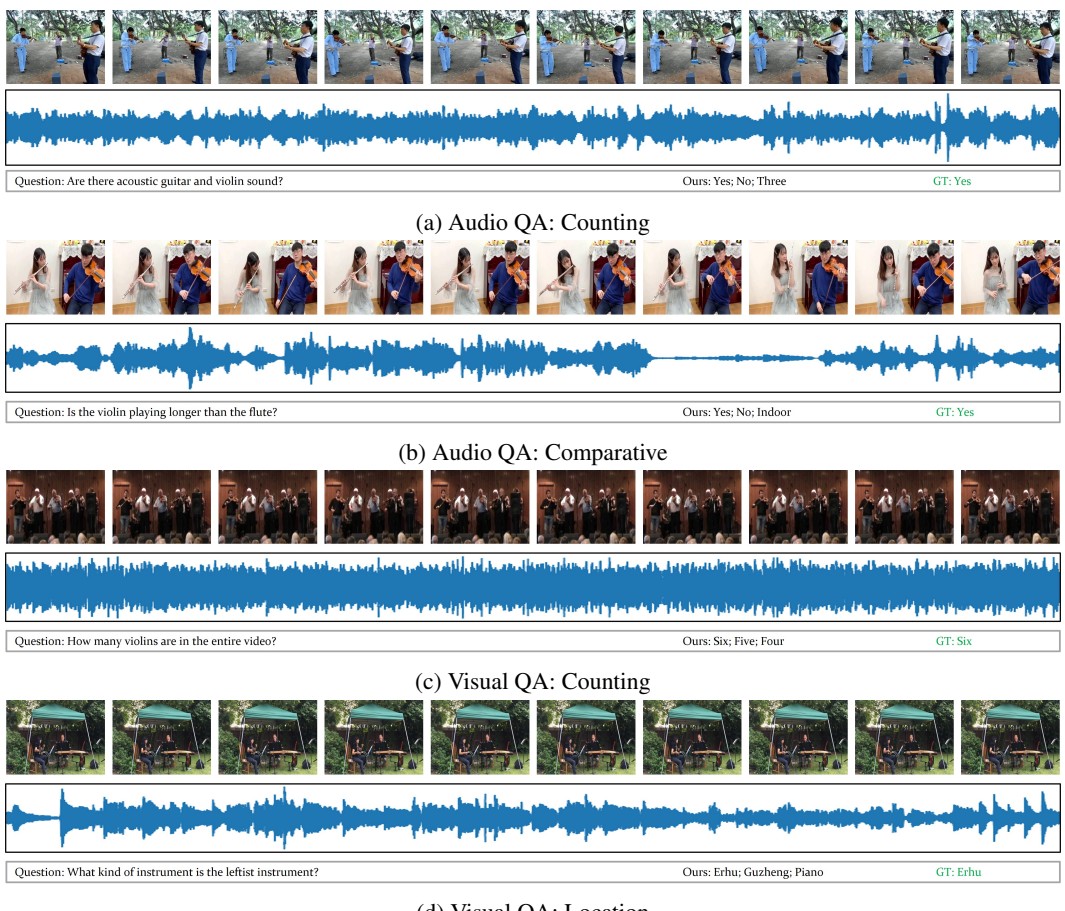

(a) Audio QA: Counting

(b) Audio QA: Comparative

(c) Visual QA: Counting

(d) Visual QA: Location

Figure 4: Prediction results of our method for different question types of Audio QA (counting, comparative) and Visual QA (counting, location) in MUSIC-AVQA (Li et al., 2022), with video ids: "00007961", "00002646", "00002454", and "00004109", respectively. We provide the top 3 answers predicted by our method in sequence.

## A.5 REPRODUCIBILITY STATEMENT

We have made every effort to ensure that the results presented in this paper are reproducible. The code will be made publicly available to facilitate replication and verification. The experimental setup, including training steps, model configurations, and hardware details, is described in detail in the paper. Additionally, the datasets used in this paper are publicly available, ensuring consistent and reproducible evaluation results.

## A.6 LLM USAGE

Large Language Models (LLMs) were used to aid in the polishing of the manuscript. Specifically, we used an LLM to assist in refining the language, improving readability, and ensuring clarity in various sections of the paper. The model helped with tasks such as sentence rephrasing, grammar checking, and enhancing the overall flow of the text. It is important to note that the LLM was not involved in the ideation, research methodology, or experimental design. All research concepts, ideas, and analyses were developed and conducted by the authors.

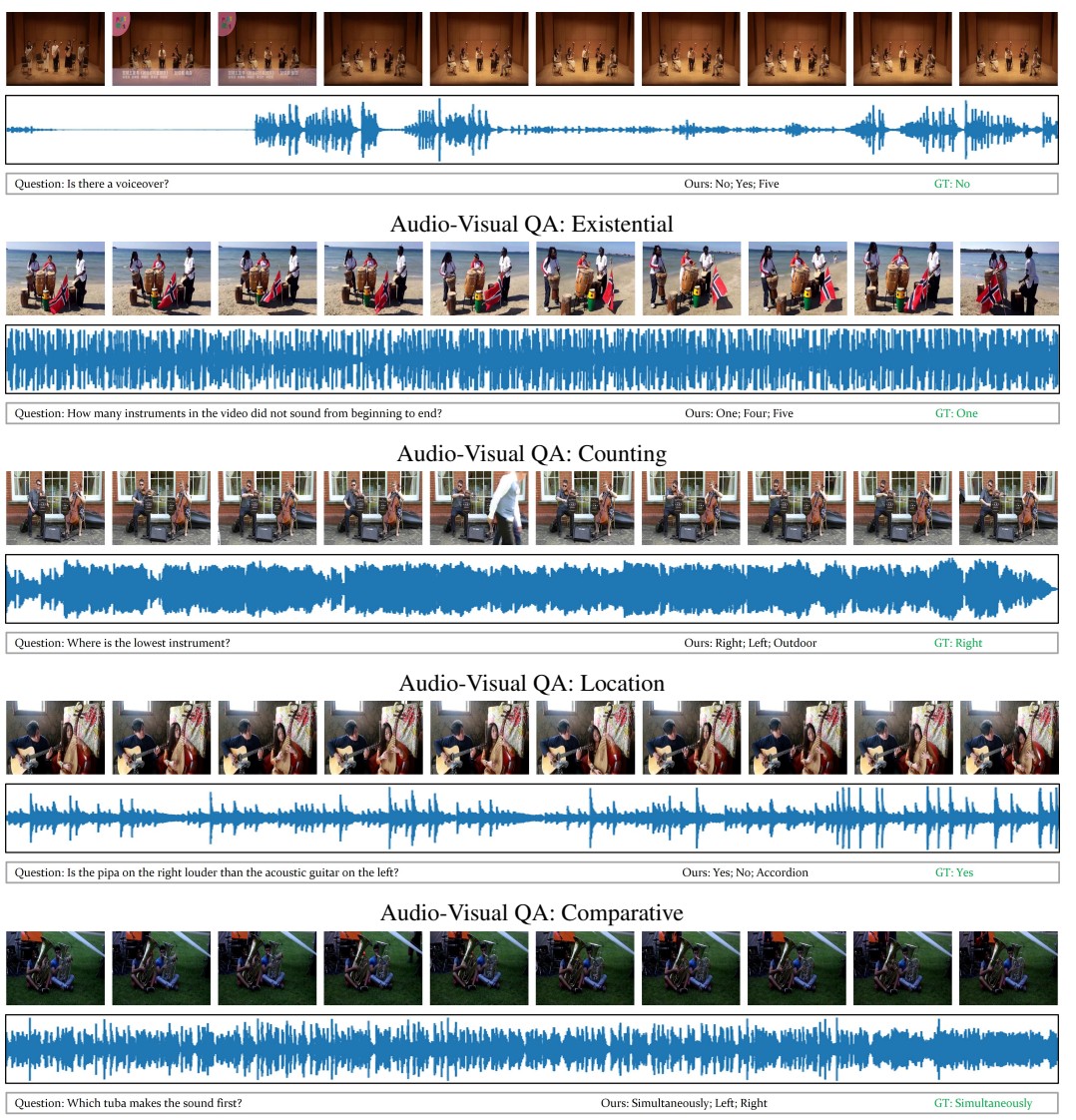

Figure 5: Prediction results of our method for different question types of Audio-Visual QA (existential, counting, location, comparative, temporal) in MUSIC-AVQA (Li et al., 2022), with video ids: "00003803", "00004995", "00008437", "00005464", and "00004026", respectively. We provide the top 3 answers predicted by our method in sequence.

