# OpenReview forum: "Query-Guided Spatial–Temporal–Frequency Interaction for Music Audio–Visual Question Answering"
_ICLR.cc/2026/Conference — ICLR 2026 Poster_

### Official Review · Reviewer_ttss · 2025-10-15

**Soundness:** 3
**Presentation:** 3
**Contribution:** 2
**Rating:** 4
**Confidence:** 4

**Summary:**

To address the audio-visual question answering task, this paper proposes a Query-guided Spatial-Temporal-Frequency (QSTar) interaction method. QStar primarily consists of the query-guided multimodal correlation module, spatial-temporal-frequency interaction module, and query context reasoning block. Experiments conducted on the MUSIC-AVQA and AVQA datasets verify the effectiveness of the proposed method.

**Strengths:**

- The discussion on the audio frequency and query context is good.
- The proposed method achieves new state-of-the-art performance on the relevant datasets.
- In general, the proposed method is described clearly.

**Weaknesses:**

- Throughout the method section, the proposed network heavily relies on prevalent self-attention and cross-attention mechanisms. The query-guided multimodal correlation module is similar to the prior method, APL. Prior work, like TSPM, etc, has already explored the utilization of question modality for early fusion. Overall, the proposed method still falls into the convention of multimodal fusion, not providing sufficient advances or improvements.

- Although the introduction of frequency makes sounds, it incorporates an additional AST backbone for feature extraction, which may make the proposed framework more complex. An analysis of the efficiency is required to justify this.

- The paper lacks a discussion or comparison of MLLMs. For the studied audio-visual question answering problem, the current omni multimodal large language models, such as Qwen2.5-Omni, Ming-Omni, and video-SALMONN2, can be used. What performance can be achieved by such omni models?

**Questions:**

- In the Introduction, the paper highlights the advantages of frequency in music scenarios, which may be suitable for the MUSIC-AVQA dataset. But the AVQA dataset contains more diverse scenarios. Would the frequency be less effective in AVQA, or would the writing in the introduction make it inconsistent for different datasets (scenarios)?
- The paper highlights the frequency by employing the AST backbone to extract features. Similar to VGGish, AST is also a general model for audio feature extraction. Why not directly use the identical AST in the temporal audio extraction?
- The query-context reasoning block utilizes the 'context-related keywords' (Line 307). How these context-related keywords are obtained?

- Figure 2 can be improved. For example, the abbreviation of several modules should be added at the end of the corresponding full module names.

---

> ### Author Response · Authors · 2025-11-20
> **Response to Reviewer ttss (1)**
>
> Thank you for recognizing our work and providing insightful comments that significantly enhance the quality of our paper.
> Here are our responses to your concerns and questions, and we hope to gain your further support.
>
>
> **Q1. Throughout the method section, the proposed network heavily relies on prevalent self-attention and cross-attention mechanisms. The query-guided multimodal correlation module is similar to the prior method, APL. Prior work, like TSPM, etc, has already explored the utilization of question modality for early fusion. Overall, the proposed method still falls into the convention of multimodal fusion, not providing sufficient advances or improvements.**
>
> A: We believe the most core and valuable aspect of our paper lies in treating AVQA as a query-guided multimodal alignment problem together with a frequency-domain reasoning perspective.
> This distinguishes QSTar from prior AVQA methods—including APL, PSTP, TSPM, and PSOT—which are primarily vision-centric.
> While our implementation uses standard transformer operations such as self- and cross-attention (as do all recent AVQA methods), these operations are not the claimed contributions.
> Below we clearly delineate the differences from other methods.
>
> APL's question-conditioned clue discovery module uses visual or audio embeddings as queries and attends to the question at a single stage.
> In contrast, our QGMC performs a three-stage process: self-enhancing, capturing, and propagating.
> Before collecting any "question-conditioned clue", QGMC uses self-enhanced word-level linguistic features as queries that attend simultaneously to frame-level visual features and audio features. This early, text-driven capturing step extracts shared semantic cues—such as instrument identity, temporal cues, and relevant appearance patterns—from both modalities before any fusion occurs.
> By aggregating these attended representations back into the residual linguistic embeddings, QGMC forms **a refined, question-aware semantic context**.
> This recursive enrichment of the linguistic representation provides a stronger guidance signal for the subsequent propagation stage, enabling more fine-grained alignment than APL's single-pass design.
>
> TSPM incorporates the question only at the final fusion and prediction stage, but introduces an auxiliary prompt (question-translation) to capture temporal segments.
> The role of the prompts is to generate attention scores for each indexing key within the video sequences.
> These scores are then subjected to a top-$k$ ranking to facilitate the selection of relevant temporal visual and audio segments.
> However, the translation-style prompts lack sufficient query specificity, often omitting critical information such as the target instruments or the focus requested by users.
> In contrast, our QSTar enables the model to attend to linguistic cues throughout the entire processing pipeline.
>
> Overall, while prior works (e.g., APL, PSTP, TSPM, PSOT) largely depend on vision-centric mechanisms such as frame selection, object detection, or motion features, our QSTar introduces a new perspective by incorporating robust **frequency-domain cues** and structuring **multimodal interactions around the question** from the beginning to the end of the pipeline.
> This allows QSTar to capture subtle distinctions in complex polyphonic scenes, which prior methods struggle to represent.
> The consistent improvements across different question types further demonstrate the value of this design.
>
> **Q2. Although the introduction of frequency makes sounds, it incorporates an additional AST backbone for feature extraction, which may make the proposed framework more complex. An analysis of the efficiency is required to justify this.**
>
> A: Thank you for raising this point.
> Although QSTar adds an extra AST-based branch, it uses a lightweight frequency-interaction module (TFI).
> As shown in the table, this leads to a modest increase of about **1.25M trainable parameters and 0.28G FLOPs** compared to QSTar without TFI, while still keeping the overall complexity comparable to or lower than several strong baselines (e.g., AVST, QA-TIGER).
> The AST branch introduces only a modest computation overhead for a **clear accuracy gain (1.57\%)**.
> This supports the efficiency of our design.
> In the revision, we have included a clear efficiency analysis to justify this in Section 4.5 and Table 4.
>
> | Method | Params (M) | FLOPs (G) | Avg Accuracy (%) |
> | :--- | :---: | :---: | :---: |
> | AVST | 18.48 | 3.19 | 71.59 |
> | LAVISH | 21.09 | - | 74.46 |
> | TSPM | 6.22 | 1.42 | 76.79 |
> | QA-TIGER | 14.51 | 2.70 | 77.62 |
> | QSTar (*w/o* TFI) | 11.95 | 2.15 | 77.41 |
> | QSTar (ours) | 13.20 | 2.43 | **78.98** |

---

> > ### Author Response · Authors · 2025-11-20
> > **Response to Reviewer ttss (2)**
> >
> > **Q3. The paper lacks a discussion or comparison of MLLMs. For the studied audio-visual question answering problem, the current omni multimodal large language models, such as Qwen2.5-Omni, Ming-Omni, and video-SALMONN2, can be used. What performance can be achieved by such omni models?**
> >
> > A: Thank you for your constructive suggestion.
> > We fully agree that demonstrating how QSTar compares to recent MLLMs would further strengthen the paper.
> >
> > To this end, we conduct additional experiments on the MUSIC-AVQA dataset and report comparisons with recent large multimodal models (MLLMs), including GPT-4o [1], Qwen2.5-Omni [2] and Ming-Omni [3].
> > Following common practice for proprietary models, these MLLMs are evaluated in a zero-shot setting due to their closed-source nature and large-scale pre-training benefits.
> > Despite their strong general multimodal reasoning abilities, **all evaluated MLLMs show clear performance ceilings** on AVQA.
> > The highly capable GPT-4o achieves only 55.72\% average accuracy while Qwen2.5-Omni (7B) and Ming-Lite-Omni achieve similar modest results at 53.83\% and 53.98\%, respectively.
> > These models struggle particularly with question types requiring fine-grained temporal and comparative reasoning in polyphonic scenes, exhibiting drops of nearly 30\% on these categories.
> > In contrast, QSTar significantly outperforms all zero-shot MLLMs across every question type.
> >
> > We further assess MLLM performance under a fine-tuning setting by comparing against VideoLLaMA2 [4], a representative open-source MLLM.
> > Even with task-specific fine-tuning, VideoLLaMA2 still **trails QSTar substantially—particularly on comparative questions**, where QSTar exceeds it by 19.36\% (Audio QA) and 7.36\% (Audio-Visual QA).
> > This performance gain highlights the specialized strength of QSTar on cross-modal reasoning within complex polyphonic scenes.
> >
> > We have added the comparison in Section 4.3 accordingly.
> >
> > | Method | Audio QA | | | Visual QA | | | Audio-Visual QA | | | | | | Avg |
> > | :--- | :---: | :---: | :---: | :---: | :---: | :---: | :---: | :---: | :---: | :---: | :---: | :---: | :---: |
> > | | Count. | Comp. | Avg | Count. | Local. | Avg | Exist. | Count. | Local. | Comp. | Temp. | Avg | |
> > | **GPT-4o** | 65.68 | 37.04 | 55.12 | 72.77 | 62.20 | 67.42 | 55.87 | 54.94 | 59.57 | 38.24 | 42.58 | 50.35 | 55.72 |
> > | **Qwen2.5-Omni** | 62.93 | 39.56 | 54.31 | 67.92 | 59.02 | 63.42 | 51.92 | 55.10 | 60.98 | 36.42 | 40.27 | 49.12 | 53.83 |
> > | **Ming-Omni** | 62.44 | 40.74 | 54.44 | 67.50 | 58.29 | 62.84 | 52.73 | 53.91 | 62.17 | 37.24 | 41.85 | 49.63 | 53.98 |
> > | **VideoLLaMa2** | 79.65 | 52.69 | 69.71 | 81.20 | 83.02 | 82.12 | 77.43 | 63.48 | 69.67 | 62.67 | 68.13 | 67.88 | 71.98 |
> > | **QSTar (ours)** | **85.64** | **72.05** | **80.63** | **83.46** | **84.90** | **84.17** | **83.81** | **79.76** | **72.72** | **70.03** | **72.38** | **75.98** | **78.98** |
> >
> > **Q4. In the Introduction, the paper highlights the advantages of frequency in music scenarios, which may be suitable for the MUSIC-AVQA dataset. But the AVQA dataset contains more diverse scenarios. Would the frequency be less effective in AVQA, or would the writing in the introduction make it inconsistent for different datasets (scenarios)?**
> >
> > A: The writing **does not introduce inconsistency** in the introduction. We first clarify the difference between MUSIC-AVQA and AVQA to explain why frequency cues play different roles across the two datasets
> >
> > - MUSIC-AVQA consists of polyphonic musical instrument performances, where each instrument exhibits rich harmonic structures, distinct overtone distributions, and timbre-specific spectral signatures.
> > Many of its questions explicitly depend on fine-grained spectral reasoning—such as identifying which instrument is sounding, determining when an instrument starts or stops, or comparing multiple performers (e.g., *Which instrument plays earlier than the pipa?*).
> > These tasks inherently require detailed frequency-domain modeling.
> >
> > - AVQA covers diverse everyday scenes (e.g., alarms, dog barking, speech, water flow, and vehicle sounds).
> > These signals are typically broadband, less structured, and often non-harmonic.
> > The corresponding questions generally focus on salient, dominant audio events (e.g., *Is the alarm ringing?*) rather than subtle harmonic differences.
> >
> > These differences arise from the nature of the audio content and the question design from the datasets, rather than from any limitation or bias in frequency modeling.
> >
> > In addition, we subsequently evaluate on AVQA to test generality beyond the musical domain, and Table 5 shows that QSTar remains competitive even when frequency cues contribute less.
> > In effect, the introduction motivates the method in the domain where frequency modeling is most impactful, while the experiments confirm that the framework still **transfers to broader AVQA scenarios**.
> > We have included more analysis in Appendix A.1.

---

> > > ### Author Response · Authors · 2025-11-20
> > > **Response to Reviewer ttss (3)**
> > >
> > > **Q5. The paper highlights the frequency by employing the AST backbone to extract features. Similar to VGGish, AST is also a general model for audio feature extraction. Why not directly use the identical AST in the temporal audio extraction?**
> > >
> > > A: AST and VGGish are indeed both general-purpose audio backbones, but in our framework they play **different and complementary roles**.
> > >
> > > - In QSTar, the temporal audio pathway is designed to capture coarse temporal dynamics for input audios.
> > > For this purpose, VGGish-style convolutional features are sufficient and, importantly, provide stable frame-level embeddings aligned to video timestamps.
> > >
> > > - In contrast, AST provides fine-grained spectral modeling with long-range attention over frequency bins, which is essential for distinguishing harmonic structures, overtone distributions, and instrument timbre (as discussed in our response to Reviewer BoFL's Q2).
> > > This is why AST features are specifically used inside the TFI module, where frequency reasoning is required.
> > >
> > > To directly address your question, we include an additional ablation where we replace the temporal VGGish pathway with AST-only features.
> > > As shown in the table, this substitution does not improve performance and in fact **slightly reduces accuracy due to poorer temporal alignment and feature redundancy**.
> > > This supports our design choice of using different audio feature extractors.
> > >
> > > | Method | Audio QA | Visual QA | Audio-Visual QA | Avg |
> > > | :--- | :---: | :---: | :---: | :---: |
> > > | *w* $F_{ast}$ | 78.96 | 83.94 | 75.12 | 78.14 |
> > > | *w* $F_a$ (ours) | **80.63** | **84.17** | **75.98** | **78.98** |
> > >
> > > **Q6. The query-context reasoning block utilizes the 'context-related keywords' (Line 307). How these context-related keywords are obtained?**
> > >
> > > A: We apologize if the source of the context-related keywords is not sufficiently clear in the main text.
> > > We actually provided a detailed analysis of how these prompts are obtained in Appendix A.2.3 and Table 8, and we summarize it here for clarity.
> > >
> > > The prompts (*type, performance duration, location, temporal sequence, loudness*) are extracted by analyzing all 33 question templates in MUSIC-AVQA and grouping them into their underlying **semantic reasoning dimensions**.
> > > As shown in Table 8, every question can be mapped to one or more of these five high-level aspects, which consistently capture what information the question requires the model to attend to.
> > > Note that we intentionally avoid using question-specific or adaptive prompts.
> > > As discussed in our response to Reviewer 92oA's Q4, such prompts risk leaking semantic hints, create fairness issues, and encourage overfitting to dataset priors.
> > >
> > > **Q7. Figure 2 can be improved. For example, the abbreviation of several modules should be added at the end of the corresponding full module names.**
> > >
> > > A: Thank you for the helpful suggestion.
> > > We have improved Figure 2 accordingly and slightly refined the overall structural presentation in the revised version.
> > >
> > > [1]. Hurst, A., Lerer, A., Goucher, A.P., Perelman, A., Ramesh, A., Clark, A., Ostrow, A.J., Welihinda, A., Hayes, A., Radford, A. and Madry, A., 2024. Gpt-4o system card. arXiv preprint arXiv:2410.21276.
> > >
> > > [2]. Xu, J., Guo, Z., He, J., Hu, H., He, T., Bai, S., Chen, K., Wang, J., Fan, Y., Dang, K. and Zhang, B., 2025. Qwen2.5-omni technical report. arXiv preprint arXiv:2503.20215.
> > >
> > > [3]. AI, I., Gong, B., Zou, C., Zheng, C., Zhou, C., Yan, C., Jin, C., Shen, C., Zheng, D., Wang, F. and Xu, F., 2025. Ming-Omni: A Unified Multimodal Model for Perception and Generation. arXiv preprint arXiv:2506.09344.
> > >
> > > [4]. Cheng, Z., Leng, S., Zhang, H., Xin, Y., Li, X., Chen, G., Zhu, Y., Zhang, W., Luo, Z., Zhao, D. and Bing, L., 2024. Videollama 2: Advancing spatial-temporal modeling and audio understanding in video-llms. arXiv preprint arXiv:2406.07476.

---

> > > > ### Comment · Reviewer_ttss · 2025-11-26
> > > >
> > > > Thanks. It is interesting to see the impact of different audio backbones (AST vs VGGish). Given that previous works do not use the AST backbone but VGGish only, it would be better to see whether simplying adding the features of AST and VGGish (eg, averge pooling can be used for addition) as the input audio features can improve the original models' performance. I am ready to increase my rating.

---

> > > > > ### Author Response · Authors · 2025-11-26
> > > > >
> > > > > Thank you for this valuable suggestion. We extend our input audio feature ablation by adding a variant that combines VGGish and AST features as the feature extraction. The results show that simply merging the two feature sets does not lead to performance improvement.
> > > > > The overall accuracy (78.65\%) remains very close to the our version (78.98\%), and the Audio QA accuracy even drops slightly (-0.8\%).
> > > > > This suggests that VGGish already provides sufficient coarse temporal cues aligned with video timestamps, while injecting additional AST features could introduce redundant or misaligned information.
> > > > > The observations further support our design choice of assigning distinct roles to the two backbones: VGGish for temporal audio pathway and AST for fine-grained frequency reasoning.
> > > > >
> > > > > | Method | Audio QA | Visual QA | Audio-Visual QA | Avg |
> > > > > | :--- | :---: | :---: | :---: | :---: |
> > > > > | *w* $F_{ast}$ | 78.96 | 83.94 | 75.12 | 78.14 |
> > > > > | *w* $F_a$ (ours) | **80.63** | 84.17 | **75.98** | **78.98** |
> > > > > | *w* $F_{ast}$ & $F_a$ | 79.83 | **84.19** | 75.65 | 78.65 |
> > > > >
> > > > > We sincerely appreciate the time and effort you dedicated to reviewing our work. Your constructive suggestions have been instrumental in refining our paper.
> > > > > We are also encouraged by your positive feedback on our responses and are glad to hear that you are ready to raise the rating!

---

> > > > > > ### Comment · Reviewer_ttss · 2025-11-27
> > > > > >
> > > > > > Thank you for the further analysis. I have no more questions to discuss. In my opinion, compared to prior methods, the paper looks like incorporating an additional 'modality' (audio extracted by AST) to highlight frequency importance in multimodal fusion. It is also interesting to see that current SOTA MLLMs (even after SFT) still do not perform very well on the stuided AVQA tasks. Overall, this paper may have its merits in the frequency utilization and experimental findings. I have revised my final rating to 6. Good luck.

---

> > > > > > > ### Author Response · Authors · 2025-11-27
> > > > > > >
> > > > > > > Dear Reviewer ttss,
> > > > > > >
> > > > > > > We greatly appreciate your recognition of our work, and thank you again for your valuable feedback and for helping us improve the paper!
> > > > > > >
> > > > > > > Best regards,
> > > > > > >
> > > > > > > The Authors

---

> > ### Comment · Reviewer_ttss · 2025-11-26
> >
> > Thank you for your reponse. So the AST features are offline extracted, and such feature extraction time is not not included and compared in total Parameters/FLOPs, right?

---

> > > ### Author Response · Authors · 2025-11-26
> > >
> > > Dear Reviewer ttss,
> > >
> > > Thank you very much for your valuable time and effort in helping us improve our work!
> > >
> > > Yes. The AST features are extracted offline, and the weights are frozen during training (as illustrated in Figure 2).
> > > Therefore, following prior works such as TSPM, we report only the trainable parameters and FLOPs of our model and compare them fairly with other AVQA methods.
> > >
> > > Best regards,
> > >
> > > The Authors

---

### Official Review · Reviewer_BoFL · 2025-11-01

**Soundness:** 3
**Presentation:** 3
**Contribution:** 3
**Rating:** 4
**Confidence:** 3

**Summary:**

The paper introduces QSTar, a novel method for Audio-Visual Question Answering (AVQA) with a strong focus on complex musical scenes. The core contribution is a multi-stage, query-guided architecture that injects textual guidance throughout the pipeline: early with a Query-Guided Multimodal Correlation (QGMC) module, in the middle via a Spatial-Temporal-Frequency Interaction (STFI) module, and finally with a Query Context Reasoning (QCR) block that uses task-aware prompts. The model explicitly enhances audio processing by incorporating frequency-domain analysis to capture timbral characteristics crucial for instrument identification. The method achieves new state-of-the-art performance on the MUSIC-AVQA benchmark, reporting an average accuracy of 78.98%. A smaller-scale evaluation on the general AVQA dataset further demonstrates the model's robustness.

**Strengths:**

- Strong Empirical Performance: The method achieves state-of-the-art results on MUSIC-AVQA, with consistent improvements across Audio, Visual, and Audio-Visual question types.
- Thorough Ablation Studies: The paper provides extensive ablations that validate the contribution of each key module (QGMC, STI, TFI, QCR) and design choice, strengthened by further controls in the supplementary material.
- The Query Context Reasoning (QCR) block implements a lightweight and reproducible prompting mechanism. It uses a fixed set of task-relevant keywords (e.g., instrument type, duration, location) derived from the dataset's question types, which are encoded and used to guide the final fusion of audio-visual features.

**Weaknesses:**

At the top: I have reviewed an earlier version of this paper. The current submission remains largely unchanged, except for minor language polishing. All major methodological and empirical concerns I previously raised remain unaddressed.
- First of all, the motivation: The paper makes broad claims that prior work treats audio as secondary and uses text late, really? How the author provide some results/examples to demonstrate it? Reader could NOT buy the motivation just by plain sentences. To summarize, the motivation does not substantiate these claims with targeted analysis or experiments on specific baselines.
- The author say AST is superior to VGGish for capturing timbral information, why? this claim is untested.
- I think the dataset scope is too narrow: Strong focus on MUSIC-AVQA; AVQA appears only as a small table; no Pano-AVQA. Limits generality claims beyond music scenes.
- My another concern drop into 'Prompting fairness': The fixed keyword prompts mirror the dataset’s question taxonomy. In the wild, question types are not given and may differ (long-tail intents, non-music scenes). The models' performance might hinges on dataset priors rather than generalizable reasoning.
- Also, given the recent progress of MLLMs (e.g., GPT-4o, Phi series) in vision-language tasks, I suggest that the authors either: (1) include a comparison between QSTar and one or more MLLM baselines, or (2) clearly explain why such comparison is not applicable and discuss the corresponding limitations.

**Questions:**

Please see the weakness part.

---

> ### Author Response · Authors · 2025-11-20
> **Response to Reviewer BoFL (1)**
>
> Thank you for recognizing the strengths of our work, particularly the strong empirical performance and thorough ablation studies, and for providing insightful comments that help further improve the paper.
> In this rebuttal, we will further address each of your comments and questions in detail. We hope the following responses clarify the changes and satisfactorily resolve your concerns.
>
> **Q1. First of all, the motivation: The paper makes broad claims that prior work treats audio as secondary and uses text late, really? How the author provide some results/examples to demonstrate it? Reader could NOT buy the motivation just by plain sentences. To summarize, the motivation does not substantiate these claims with targeted analysis or experiments on specific baselines.**
>
> A: We agree that the motivation should be supported with clearer evidence, and we appreciate the opportunity to clarify this.
> Although other reviewers (e.g., 92oA and VpPp) explicitly recognize our paper as **"well-motivated"** and find the introduction of frequency-level interaction **"interesting and meaningful"**, we understand the need to make the motivation more explicit.
> Below we provide targeted analysis and references to support the two core claims.
>
> 1. Audio treated as secondary in prior AVQA models.
>
> Many representative AVQA methods integrate audio after visual processing or use it primarily for temporal alignment.
> For instance, in PSTP, audio is only used to guide temporal frame selection, while semantic reasoning is performed mainly on visual features.
> In APL, audio merely matches pre-detected visual objects, but the semantic reasoning depends heavily on the detectors.
> In PSOT, audio is used only to select sound-driven patches that are then aggregated into a largely vision-centric pipeline.
> To visually illustrate the limitation of such vision-centric approaches, we provide a **concrete example** in Figure 1.
> When subtle motions of the flutist cannot be reliably captured in RGB frames, the absence of corresponding audio patterns clearly signals that the flute is not continuously performing, where information unavailable to purely visual reasoning.
> To further substantiate our observation, we conduct **an ablation** where we remove our frequency-aware audio modeling.
> As shown in Table 2, performance drops by 1.57\% on average, confirming the importance of richer audio modeling for AVQA.
>
> 2. Late incorporation of the question in prior models.
>
> Many widely used baselines inject the question only after audio and visual integration.
> For instance, AVST first performs spatial grounding using audio and visual features, and only then applies temporal reasoning with question features.
> TSPM incorporates the question only at the final fusion and prediction stage (with an auxiliary prompt capturing temporal segments).
> Similarly, we validate the importance of early query guidance through **an ablation study in Table 3**.
> Removing early query integration leads to a 1.05\% performance decline, demonstrating that introducing textual cues at the beginning benefits cross-modal alignment for AVQA.
>
> To make the motivation clearer and more convincing, we have integrated the above evidence and analyses into the revised manuscript (Section 1 and 2).
>
> **Q2. The author say AST is superior to VGGish for capturing timbral information, why? this claim is untested.**
>
> A: Thank you for pointing this out.
> Timbre is defined by a sound's unique spectral characteristics, which are fundamentally frequency-domain patterns.
> These include the specific harmonic structure, the spectral envelope, and the distribution of overtone amplitudes [1-2].
> Our statement that AST is better suited for modeling timbral information than VGGish is based on **their architectural differences, supported by findings in prior audio research**.
>
> - AST operates on spectrogram patches using a ViT-style transformer, enabling it to capture long-range dependencies across frequency bins.
> This makes it effective for distinguishing harmonic structures and overtone patterns that characterize timbre.
> VGGish is based on VGG-style convolution layers and its local receptive fields are less effective for modeling fine spectral structures that differentiate similar-pitch instruments (e.g., violin vs. erhu).
>
> - Research such as AST, PaSST [3], and HTS-AT [4] show that transformer-based audio models outperform VGGish on tasks requiring detailed spectral or harmonic discrimination (e.g., instrument classification, environmental sound tagging).
>
> Additionally, **ablation of the AST-derived frequency modeling** within the TFI module results in a 1.37\% reduction in overall performance (in Table 7), underscoring the importance of frequency-domain cues from AST.
>
> To avoid overstating the claim, we have revised the paper to state that AST is better suited for timbre modeling in frequency domain, and included relevant citations in Section 3.3.

---

> > ### Author Response · Authors · 2025-11-20
> > **Response to Reviewer BoFL (2)**
> >
> > **Q3. I think the dataset scope is too narrow: Strong focus on MUSIC-AVQA; AVQA appears only as a small table; no Pano-AVQA. Limits generality claims beyond music scenes.**
> >
> > A: The title of our work, "Query-Guided Spatial-Temporal-Frequency Interaction for Music Audio-Visual Question Answering", reflects that the method is specifically designed and optimized for music-related AVQA, where **frequency-domain modeling and polyphonic interactions** are central.
> > Therefore, our primary focus on MUSIC-AVQA is intentional grounded in several practical and methodological considerations.
> >
> > 1. MUSIC-AVQA is the most widely adopted benchmark for AVQA research.
> >
> > Nearly all recent AVQA methods such as AVST, PSTP, APL, PSOT, and QA-TIGER, use MUSIC-AVQA as their **primary evaluation dataset** because it (1) offers the most comprehensive annotation for audio-visual reasoning; (2) includes fine-grained question types; (3) involves challenging polyphonic scenes that are ideal for studying frequency-domain modeling.
> > Thus, evaluating on MUSIC-AVQA aligns with current mainstream practice in the field.
> >
> > 2. We additionally evaluate on AVQA to demonstrate generality beyond musical scenes.
> >
> > To address cross-domain applicability, we also report results on the AVQA dataset, which covers broader environmental and conversational scenarios.
> > This dataset covers diverse everyday scenes (e.g., alarms, dog barking, speech, water flow, and vehicle sounds).
> > These signals are typically broadband, less structured, and often non-harmonic.
> > The corresponding questions generally focus on **salient, dominant audio events** (e.g., *Is the alarm ringing?*) rather than subtle harmonic differences.
> >
> > We acknowledge the compact presentation of the results in Table 5.
> > This format is necessitated by the limited number of existing methods that report performance on this dataset, and the restriction of their reported metrics solely to overall accuracy.
> > To ensure **direct comparability and mitigate potential implementation discrepancies**, we thus follow the exact reporting format established by prior methods.
> > From this table, QSTar still achieves **consistent improvements** over prior work, indicating that the proposed modules (query guidance, multimodal correlation, STFI, QCR) are not restricted to music-specific settings.
> > We have included more analysis in Appendix A.1.
> >
> > 3. Pano-AVQA is not directly compatible with the standard AVQA setting.
> >
> > Pano-AVQA is based on 360° panoramic videos with ambisonic spatial audio, where reasoning depends heavily on **panoramic geometry and direction-of-arrival cues**.
> > This requires specialized visual backbones and spatial-audio encoding pipelines that are not supported by VGGish/AST or other AVQA models.
> > Although such evaluation would **not be comparable or fair**, we still test QSTar on Pano-AVQA and compare the results with available methods (AVSD, LXMERT [5], LAViT [6]) for your question; as expected, performance is limited without ambisonic-specific modeling (see the table below).
> > This explains why recent AVQA works (e.g., APL, PSTP, PSOT, QA-TIGER) also typically do not evaluate on Pano-AVQA.
> > Thus, we follow these methods and do not report related results.
> >
> > | Method | Audio-Visual Accuracy |
> > | :--- | :---: |
> > | AVSD | 20.10 |
> > | LXMERT | 49.12 |
> > | LAViT | **51.25** |
> > | QSTar (ours) | 49.73 |
> >
> > Overall, our goal is not to claim universal generality, but to improve multimodal reasoning with frequency-aware modeling.
> > MUSIC-AVQA provides a challenging testbed for this purpose, and the additional AVQA results demonstrate that our design generalizes beyond music scenes.
> > We agree that broader evaluation is valuable, and we will be glad to explore the effectiveness of our approach on suitable datasets in future work (Appendix A.4).
> >
> > **Q4. My another concern drop into 'Prompting fairness': The fixed keyword prompts mirror the dataset’s question taxonomy. In the wild, question types are not given and may differ (long-tail intents, non-music scenes). The models' performance might hinges on dataset priors rather than generalizable reasoning.**
> >
> > A: We appreciate the opportunity to clarify why our prompting strategy is designed this way and how it remains fair and generalizable beyond the MUSIC-AVQA taxonomy. Here follow three points.
> >
> > 1. The prompts do not encode dataset priors.
> >
> > The five prompts we use, *type, performance duration, location, temporal sequence, loudness*, are not one-hot question labels or dataset-specific templates.
> > They are **high-level semantic aspects that commonly appear in any audio-visual reasoning tasks**: identifying what is sounding or appearing, when an event occurs, where it occurs, in what order, and with what intensity.
> > These prompts are not tied to MUSIC-AVQA and do not create shortcuts to the answer space.
> > Instead, they define broad reasoning dimensions, which we call query context in this work.

---

> > > ### Author Response · Authors · 2025-11-20
> > > **Response to Reviewer BoFL (3)**
> > >
> > > 2. Prompts are intentionally generic, not specific for each question.
> > >
> > > We use unified prompts rather than question-specific or generative prompts.
> > > This strict idea is essential to **prevent semantic leakage and avoid providing models with unfair inferential advantages**.
> > > For instance, consider the question: *Can you see a man performing a piano?*
> > > A specific prompt, such as asking the model only about "existence" or the object "piano," would immediately restrict the required reasoning field.
> > > This leakage bypasses the need for the model to correctly identify all relevant objects and actions, thereby creating a semantic hint that boosts performance.
> > >
> > > Likewise, **generative prompts**, which we tested in the original submission (translation- and caption-based) and extend in this rebuttal to constrained generative descriptions (see our detailed response to Reviewer 92oA's Q4), tend to introduce semantic noise, drift away from the actual question, or bias the model toward linguistic priors rather than audio–visual content.
> > > These observations further support our choice of a unified, domain-agnostic prompting strategy that does not rely on knowing the question type beforehand.
> > >
> > > 3. These prompts do not determine the model’s reasoning, but provide query context.
> > >
> > > The prompts serve as contextual anchors, not instructions.
> > > They do not specify which instrument to focus on or what the answer might be or what reasoning steps to follow.
> > > They act as **contextual cues** after the main cross-modal alignment has already been performed by QGMC and STFI.
> > > This is reflected in the relatively **modest but consistent performance gains** (0.79\% on MUSIC-AVQA and 0.3\% on AVQA) compared to the other proposed modules.
> > > These numbers indicate that QCR contributes helpful refinement without dominating the behavior of the model or embedding dataset priors.
> > >
> > > We have revised the manuscript to explain why unified prompts improve fairness and generality, and incorporated the additional experiments comparing alternative generative prompting strategies in Appendix A.2.3.
> > >
> > > **Q5. Also, given the recent progress of MLLMs (e.g., GPT-4o, Phi series) in vision-language tasks, I suggest that the authors either: (1) include a comparison between QSTar and one or more MLLM baselines, or (2) clearly explain why such comparison is not applicable and discuss the corresponding limitations.**
> > >
> > > A: Thank you for your constructive suggestion.
> > > We fully agree that demonstrating how the proposed QSTar compares to recent MLLMs would further strengthen the paper.
> > >
> > > To this end, we conduct additional experiments on the MUSIC-AVQA dataset and report comparisons with recent large multimodal models (MLLMs), including GPT-4o [7], Qwen2.5-Omni [8], and Ming-Omni [9].
> > > Following common practice for proprietary models, these MLLMs are evaluated in a zero-shot setting due to their closed-source nature and large-scale pre-training benefits.
> > > Despite their strong general multimodal reasoning abilities, **all evaluated MLLMs show clear performance ceilings** on AVQA.
> > > The highly capable GPT-4o achieves only 55.72\% average accuracy while Qwen2.5-Omni (7B) and Ming-Lite-Omni achieve similar modest results at 53.83\% and 53.98\%, respectively.
> > > These models struggle particularly with question types requiring fine-grained temporal and comparative reasoning in polyphonic scenes, exhibiting drops of nearly 30\% on these categories.
> > > In contrast, QSTar significantly outperforms all zero-shot MLLMs across every question type.
> > >
> > > We further assess MLLM performance under a fine-tuning setting by comparing against VideoLLaMA2 [10], a representative open-source MLLM.
> > > Even with task-specific fine-tuning, VideoLLaMA2 **still trails QSTar substantially**—particularly on comparative questions, where QSTar exceeds it by 19.36\% (Audio QA) and 7.36\% (Audio-Visual QA).
> > > This performance gain highlights the specialized strength of QSTar on cross-modal reasoning within complex polyphonic scenes.
> > >
> > > In summary, QSTar delivers stronger multimodal integration and superior comparative and temporal in complex music scenes, outperforming both zero-shot and fine-tuned MLLMs across all question types for AVQA.
> > > We have added the comparison in Section 4.3 accordingly.

---

> > > > ### Author Response · Authors · 2025-11-20
> > > > **Response to Reviewer BoFL (4)**
> > > >
> > > > | Method | Audio QA | | | Visual QA | | | Audio-Visual QA | | | | | | Avg |
> > > > | :--- | :---: | :---: | :---: | :---: | :---: | :---: | :---: | :---: | :---: | :---: | :---: | :---: | :---: |
> > > > | | Count. | Comp. | Avg | Count. | Local. | Avg | Exist. | Count. | Local. | Comp. | Temp. | Avg | |
> > > > | **GPT-4o** | 65.68 | 37.04 | 55.12 | 72.77 | 62.20 | 67.42 | 55.87 | 54.94 | 59.57 | 38.24 | 42.58 | 50.35 | 55.72 |
> > > > | **Qwen2.5-Omni** | 62.93 | 39.56 | 54.31 | 67.92 | 59.02 | 63.42 | 51.92 | 55.10 | 60.98 | 36.42 | 40.27 | 49.12 | 53.83 |
> > > > | **Ming-Omni** | 62.44 | 40.74 | 54.44 | 67.50 | 58.29 | 62.84 | 52.73 | 53.91 | 62.17 | 37.24 | 41.85 | 49.63 | 53.98 |
> > > > | **VideoLLaMA2** | 79.65 | 52.69 | 69.71 | 81.20 | 83.02 | 82.12 | 77.43 | 63.48 | 69.67 | 62.67 | 68.13 | 67.88 | 71.98 |
> > > > | **QSTar (ours)** | **85.64** | **72.05** | **80.63** | **83.46** | **84.90** | **84.17** | **83.81** | **79.76** | **72.72** | **70.03** | **72.38** | **75.98** | **78.98** |
> > > >
> > > >
> > > > [1]. Agostini, G., Longari, M. and Pollastri, E., 2003. Musical instrument timbres classification with spectral features. EURASIP Journal on Advances in Signal Processing, 2003(1), p.943279.
> > > >
> > > > [2]. Zhao, H., Gan, C., Rouditchenko, A., Vondrick, C., McDermott, J. and Torralba, A., 2018. The sound of pixels. In Proceedings of the European conference on computer vision (ECCV) (pp. 570-586).
> > > >
> > > > [3]. Koutini, K., Schlüter, J., Eghbal-Zadeh, H. and Widmer, G., 2021. Efficient training of audio transformers with patchout. arXiv preprint arXiv:2110.05069.
> > > >
> > > > [4]. Chen, K., Du, X., Zhu, B., Ma, Z., Berg-Kirkpatrick, T. and Dubnov, S., 2022. Hts-at: A hierarchical token-semantic audio transformer for sound classification and detection. In IEEE International Conference on Acoustics, Speech and Signal Processing (ICASSP) (pp. 646-650).
> > > >
> > > > [5]. Tan, H. and Bansal, M., 2019. LXMERT: Learning Cross-Modality Encoder Representations from Transformers. In Proceedings of the 2019 Conference on Empirical Methods in Natural Language Processing and the 9th International Joint Conference on Natural Language Processing (EMNLP-IJCNLP) (pp. 5100-5111).
> > > >
> > > > [6]. Yun, H., Yu, Y., Yang, W., Lee, K. and Kim, G., 2021. Pano-avqa: Grounded audio-visual question answering on 360deg videos. In Proceedings of the IEEE/CVF International Conference on Computer Vision (pp. 2031-2041).
> > > >
> > > > [7]. Hurst, A., Lerer, A., Goucher, A.P., Perelman, A., Ramesh, A., Clark, A., Ostrow, A.J., Welihinda, A., Hayes, A., Radford, A. and Madry, A., 2024. Gpt-4o system card. arXiv preprint arXiv:2410.21276.
> > > >
> > > > [8]. Xu, J., Guo, Z., He, J., Hu, H., He, T., Bai, S., Chen, K., Wang, J., Fan, Y., Dang, K. and Zhang, B., 2025. Qwen2.5-omni technical report. arXiv preprint arXiv:2503.20215.
> > > >
> > > > [9]. AI, I., Gong, B., Zou, C., Zheng, C., Zhou, C., Yan, C., Jin, C., Shen, C., Zheng, D., Wang, F. and Xu, F., 2025. Ming-Omni: A Unified Multimodal Model for Perception and Generation. arXiv preprint arXiv:2506.09344.
> > > >
> > > > [10]. Cheng, Z., Leng, S., Zhang, H., Xin, Y., Li, X., Chen, G., Zhu, Y., Zhang, W., Luo, Z., Zhao, D. and Bing, L., 2024. Videollama 2: Advancing spatial-temporal modeling and audio understanding in video-llms. arXiv preprint arXiv:2406.07476.

---

> ### Author Response · Authors · 2025-11-25
>
> Dear Reviewer BoFL,
>
> We sincerely appreciate your thoughtful consideration of our rebuttals. Your insightful questions have deepened our work.
>
> We are truly grateful for your engagement and for recognizing our work's contributions by raising the score to 6. Thank you very much for your valuable time and effort in helping us improve our work!
>
> Best regards,
>
> The Authors

---

### Official Review · Reviewer_VpPp · 2025-11-01

**Soundness:** 3
**Presentation:** 2
**Contribution:** 3
**Rating:** 6
**Confidence:** 4

**Summary:**

This paper proposes QSTar, a Query-guided Spatial–Temporal–Frequency Interaction framework for Audio–Visual Question Answering task, with a focus on music-related datasets. The key idea is to enhance multimodal reasoning by introducing frequency-domain analysis alongside spatial and temporal interactions. The method introduces three main components: Query-Guided Multimodal Correlation (QGMC) module for early query-conditioned feature alignment; Spatial–Temporal–Frequency Interaction (STFI) module for fine-grained multimodal fusion; and Query Context Reasoning (QCR) block inspired by prompt-based context modeling.
Experiments on MUSIC-AVQA show that QSTar achieves competitive results.

**Strengths:**

- It is interesting and meaningful to introduce frequency-level interaction in AVQA tasks. This idea is natural given the acoustic characteristics of music-related videos and provides a new perspective besides conventional spatial–temporal fusion.
- The overall framework is well structured, clearly integrating linguistic guidance across multiple stages of the model.
- The implementation details and experimental setup are well described, and the ablation studies comprehensively verify the contribution of individual modules.

**Weaknesses:**

- The experimental analysis is rather shallow. Most evaluations focus only on accuracy numbers. More in-depth discussion on how frequency-level cues contribute to specific question types or modalities would strengthen the claim. For example, what kinds of questions benefit most from frequency reasoning and why?
- The improvements over recent strong baselines (e.g., QA-TIGER) are relatively modest—about 1–2% overall—and mainly appear on Audio QA and temporal AVQA subsets. There are no gains in Visual QA. This raises questions about the generality of the proposed design beyond audio-dominant scenarios.
- The ablation results show only slight differences between “with QCR” and “without QCR,” suggesting that the reasoning block contributes limited additional value.
- The paper would benefit from more qualitative or case-level studies, such as visualizing which frequency bands or audio/video segments are attended to under different query types. This would make the contribution of “frequency-level interaction” more convincing.

**Questions:**

Please check the above section.

---

> ### Author Response · Authors · 2025-11-20
> **Response to Reviewer VpPp (1)**
>
> We sincerely appreciate your thorough reading of our paper and your many constructive comments.
> Below, we address each of your concerns and clarify potential misunderstandings.
>
> **Q1. The experimental analysis is rather shallow. Most evaluations focus only on accuracy numbers. More in-depth discussion on how frequency-level cues contribute to specific question types or modalities would strengthen the claim. For example, what kinds of questions benefit most from frequency reasoning and why?**
>
> A: Thank you for your constructive suggestion.
> We have incorporated these analysis accordingly in Section 4.3 and 4.4.
>
> Audio QA and Audio-Visual QA exhibit the most significant performance gains from our frequency-enhanced TFI module.
> When the TFI module is removed, accuracy drops by 2.42\% and 1.59\%, respectively, indicating that questions requiring recognition of which instrument is sounding or when it becomes active rely heavily on frequency-domain cues.
> The improvement is particularly notable for **comparative and temporal** questions, where reasoning depends on detecting onset/offset transitions that are often imperceptible in RGB frames.
> For example, in Audio QA, our method achieves a 4.2\% gain over QA-TIGER on comparative questions.
> In Audio-Visual QA, comparative and temporal questions improve by 6.09\% and 2.79\%, respectively, compared with QA-TIGER.
> These gains arise because frequency energy patterns provide precise indicators of instrument activity, especially for **polyphonic scenes**.
> Unlike visual cues, which may be subtle, occluded, or ambiguous, frequency features clearly reflect when an instrument starts, stops, or changes intensity.
> This is especially beneficial for rare or visually ambiguous instruments such as erhu or pipa, whose distinctive spectral signatures enable the model to distinguish them even when visual evidence is weak (see our detailed response to Reviewer 92oA' Q3).
>
> **Q2. The improvements over recent strong baselines (e.g., QA-TIGER) are relatively modest—about 1–2\% overall—and mainly appear on Audio QA and temporal AVQA subsets. There are no gains in Visual QA. This raises questions about the generality of the proposed design beyond audio-dominant scenarios.**
>
> A: Thank you for this thoughtful observation.
> We agree that the improvements over QA-TIGER are more pronounced on Audio QA and temporal, comparative Audio-Visual QA.
> This is consistent with **both the design intent of QSTar and the challenges of MUSIC-AVQA**.
> Our method focuses on frequency-aware modeling and query-guided cross-modal alignment, which naturally benefits tasks where audio provides essential discriminative cues.
> In contrast, Visual QA relies entirely on appearance-based recognition and does not require audio information by definition.
>
> Despite not using object detectors or specialized visual perception modules, QSTar remains highly competitive on Visual QA, trailing QA-TIGER by only 0.97\%.
> The primary performance gap in favor of QA-TIGER appears in location-related questions, reflecting the gains provided by its complex MoE-based spatial recognition enhancement.
> However, QSTar still **significantly outperforms other vision-centric methods such as APL (+4.48\%) and PSOT (+4.1\%)**.
> This demonstrates that our query-guided multimodal design preserves strong visual reasoning.
> Additionally, the significant gains on Audio-Visual questions, highlight the strength of our method in **the most challenging aspects of the AVQA task**, clearly distinguishing it from standard VQA.
> We acknowledge that incorporating stronger visual priors (e.g., object detectors or motion features) could further enhance performance on Visual QA, and we have included this as a promising direction for future work (Appendix A.4).
>
> **Q3. The ablation results show only slight differences between “with QCR” and “without QCR,” suggesting that the reasoning block contributes limited additional value.**
>
> A: We agree that the improvement from QCR (0.79\% on average) is smaller than those contributed by other modules.
> This is expected, as QCR is designed as a refinement step applied after the major multimodal alignment has already been established by QGMC and STFI.
> Its role is to **provide linguistic context, rather than to reshape audio-visual representations at scale**.
> Although modest in magnitude, QCR yields consistent improvements across all question categories, indicating that it provides stable complementary information rather than isolated or sporadic gains.
> This consistency supports the value of including QCR as an effective enhancement to the overall pipeline.
>
> **Q4. The paper would benefit from more qualitative or case-level studies, such as visualizing which frequency bands or audio/video segments are attended to under different query types. This would make the contribution of “frequency-level interaction” more convincing.**
>
> A: Thank you for this insightful suggestion. We have provided more qualitative results in Appendix A.3.

---

> > ### Comment · Reviewer_VpPp · 2025-11-26
> >
> > Thank you for the detailed response and further discussion. Most of my concerns are solved, and I will keep my positive score.

---

> > > ### Author Response · Authors · 2025-11-26
> > >
> > > Dear Reviewer VpPp,
> > >
> > > Thank you for taking the time to read through our response and providing positive feedback.
> > >
> > > Best regards,
> > >
> > > The Authors

---

### Official Review · Reviewer_92oA · 2025-11-01

**Soundness:** 3
**Presentation:** 3
**Contribution:** 3
**Rating:** 6
**Confidence:** 4

**Summary:**

The submitted manuscript addresses the problem that previous studies in AVQA task often overlooked the importance of the audio modality. To tackle this issue, the authors propose a Query-guided Spatial-Temporal-Frequency framework (QSTar), which effectively integrates question-guided cues with the distinctive frequency-domain characteristics and spatio-temporal perception of audio signals to enhance audio-visual understanding. Experiments conducted on relevant datasets demonstrate the effectiveness of the proposed method. Overall, the framework exhibits a clear novelty.

**Strengths:**

1. The problem addressed in the manuscript is clearly defined and well-motivated. By focusing on frequency-domain information from the audio perspective, the work presents a distinctive approach that draws attention.
2. The proposed QSTar framework shows a degree of originality and has been extensively validated on multiple datasets, confirming its effectiveness.
3. The writing is clear and well-structured, making the paper easy to read and understand.

**Weaknesses:**

1. Figure 1 effectively illustrates how frequency-domain cues assist in detecting instrument activities that purely spatial or temporal reasoning may miss. However, can similar improvements be observed for fine-grained visual understanding?
2. How does the model handle off-screen audio sources, such as when the sound of an instrument exists in the audio but the corresponding instrument does not appear in the video?
3. Considering the limited spatial supervision for audio-visual correspondence, how does the method effectively associate audio and visual cues, especially for rare instruments such as the suona or erhu?
4. Although the motivation for QCR originates from the analysis of the MUSIC-AVQA dataset, the paper does not discuss in depth the specific prompt forms or their performance compared with more dynamic or generative alternatives, raising some concerns about the scalability of the approach.
5. Some writing suggestions include avoiding widowed words at the end of paragraphs and adding references for the comparative methods listed in the tables.

**Questions:**

My main questions are reflected in the Weaknesses Section.

---

> ### Author Response · Authors · 2025-11-20
> **Response to Reviewer 92oA (1)**
>
> We would like to thank you for the thoughtful and constructive feedback, as well as the positive assessment regarding the clarity, motivation, novelty, and effectiveness of our work. Below we provide our responses to your questions in detail.
>
> **Q1: Figure 1 effectively illustrates how frequency-domain cues assist in detecting instrument activities that purely spatial or temporal reasoning may miss. However, can similar improvements be observed for fine-grained visual understanding?**
>
> A: In this work, our frequency-domain modeling is applied only to the audio modality, as audio frequency patterns (timbre, harmonics, onset energy) provide discriminative cues that are unavailable in visual frames.
> We **do not perform explicit visual frequency-domain processing** (e.g., high-pass/low-pass filtering or DCT-based feature extraction).
> This is a deliberate design choice for two reasons:
>
> - Visual frequency filtering can distort object shapes, textures, and motion cues, which are critical in music-performance scenarios where instruments are mostly rigid and motion is relatively subtle.
>
> - Although the frequency-domain cues in our model primarily enhance audio representation, they also contribute to fine-grained visual understanding indirectly by enabling tighter multimodal alignment in the later fusion stage. Furthermore, the patch-level visual features help strengthen visual representations under query guidance, enabling the model to better localize sounding regions.
> This benefit is reflected in the improved performance reported in Table 1, where our method surpasses the vision-focused APL method by 4.48\% and 5.02\% on Visual QA and Audio-Visual QA, respectively.
>
> Nonetheless, we agree that explicit visual frequency decomposition is an interesting future direction. We have clarified this in the revision and incorporated a discussion in the future work section (Appendix A.4).
>
> **Q2. How does the model handle off-screen audio sources, such as when the sound of an instrument exists in the audio but the corresponding instrument does not appear in the video?**
>
> A: Our model relies on **both audio and visual cues** for prediction.
> In cases where an instrument is audible but not visible in a specific frame, the model primarily leverages the audio pathway while still **propagating shared temporal and instrument-related information** through the query-guided modules.
> Although an instrument may be absent in some frames, MUSIC-AVQA videos typically contain other frames where the instrument appears, providing sufficient visual context for multimodal reasoning.
>
> The dataset includes **very few** instances where an instrument sounds without being visible at all, so the model has limited exposure to true off-screen audio events.
> In such cases, spatial or location-dependent questions (e.g., *Where is the violin?* or *Is the drum on the left?*) may be ambiguous because no visual grounding is available.
> It is also worth noting that the audio features in our framework are not required to correspond to individual objects.
> Instead, the segments most relevant to the query are emphasized through our audio refinement process across the entire pipeline.
> This design allows the model to focus on the correct audio cues even when the visual evidence is missing or partially occluded.
> The **left example in Figure 3 (a)** also verifies the effectiveness of QSTar in capturing the missing cello in the middle frame.
>
> **Q3. Considering the limited spatial supervision for audio-visual correspondence, how does the method effectively associate audio and visual cues, especially for rare instruments such as the suona or erhu?**
>
> A: Spatial supervision (e.g., bounding boxes or masks) is not required in standard VQA settings.
> Similarly, our method learns audio–visual correspondence without dense spatial annotations, and its **ability to associate multimodal cues arises from several design components**:
>
> - The proposed QGMC module aligns audio and visual features only along dimensions relevant to the question.
> This targeted interaction enables the model to attend to the correct audio-visual segments even in the absence of explicit spatial supervision.
>
> - For rare instruments such as suona or erhu, visual cues are often ambiguous due to limited training samples.
> However, their audio signatures (timbre, harmonic structure, onset patterns) are often highly discriminative.
> Our TFI module, built on AST-based frequency-aware features, captures these discriminative patterns and later fuses them with visual cues, allowing the model to correctly identify and associate rare instruments even with weak visual evidence.
>
> - The STI module operates on patch-level visual features, which provide fine-grained spatial information.
> Through multimodal fusion, our QSTar gradually learns consistent co-occurrence between specific audio signatures and relevant visual features, forming reliable audio--visual associations without explicit spatial supervision.

---

> > ### Author Response · Authors · 2025-11-20
> > **Response to Reviewer 92oA (2)**
> >
> > We further include a table of results on questions involving **rare instruments (e.g., suona, pipa and erhu)**.
> > As shown, our method surpasses APL, which leverages object information from a pre-trained object detector (though not full spatial supervision), demonstrating the effectiveness of our query-guided multimodal learning strategy for low-frequency instrument categories.
> > Moreover, although rare-instrument questions naturally lead to lower accuracy relative to the full set, the reduction (numbers in parentheses with $\downarrow$) is significantly smaller than that of APL.
> > This indicates that QSTar maintains stronger and more stable audio–visual associations, even when visual cues or sample counts are limited.
> >
> > | | Audio QA | Visual QA | Audio-Visual QA | Avg |
> > | :--- | :---: | :---: | :---: | :---: |
> > | **\# of Samples** | 190 | 168 | 249 | - |
> > | **APL** | 76.32 (1.77$\downarrow$) | 74.40 (5.29$\downarrow$) | 68.27 (2.69$\downarrow$) | 72.49 (2.04$\downarrow$) |
> > | **QSTar (ours)** | **79.47** (1.16$\downarrow$) | **82.14** (2.03$\downarrow$) | **73.90** (2.08$\downarrow$) | **77.92** (1.06$\downarrow$) |
> >
> > **Q4. Although the motivation for QCR originates from the analysis of the MUSIC-AVQA dataset, the paper does not discuss in depth the specific prompt forms or their performance compared with more dynamic or generative alternatives, raising some concerns about the scalability of the approach.**
> >
> > A: Thank you for pointing this out.
> > Our design of QCR is motivated by the observation that MUSIC-AVQA questions consistently revolve around a small set of interpretable semantic factors, as discussed in Appendix A.2.3 and Table 8.
> >
> > Grounded in this structure, we adopt concise and interpretable prompts that explicitly correspond to these reasoning dimensions.
> > The unified prompt set is: ***type; performance duration; location; temporal sequence; loudness***.
> > For example, under the *type* aspect, the underlying focus may concern instrument category or audio identity.
> > However, we **do not** craft question-specific prompts or explanations.
> > We intentionally use a unified prompt formulation for two reasons:
> >
> > - **Avoiding prompt-answer leakage**. Using tailored prompts per question risks encoding partial answers or shortcuts. Unified prompts ensure the model must rely on audio, visual, and textual cues during reasoning.
> >
> > - **Ensuring scalability and stability**. A consistent, domain-agnostic prompt set scales naturally to new question types and datasets, while also providing stable semantic anchors for multimodal alignment.
> > We also report results on the AVQA dataset, both with and without prompts (in Appendix A.1 and Table 5), which further demonstrates the scalability and stability of our design.
> >
> > It is also important to note that our prompts are not **instructions**.
> > They function as internal reasoning cues, embedded in the feature-processing pipeline prior to multimodal fusion, aiming at stabilizing alignment rather than guiding output generation.
> > As shown in Table 9, we already compared our unified prompts against two alternatives: (1) **translation-based prompts**, and (2) **caption-based prompts** (descriptive generative text).
> > Both methods introduce semantic drift and noise, and underperform.
> >
> > To further address your question about scalability to more dynamic or generative strategies, we now include an additional comparison using **attribute-expansion prompts**.
> > We use the following generative-style prompt template: *A detailed description of instruments, their location, appearance, and actions that help answer the $<$question$>$.*
> > The produced long-form natural-language descriptions by GPT-4o [1] are encoded using the same pretrained text encoder.
> > Although these prompts provide more constrained descriptions than free-form captions, they still introduce substantial irrelevant details. Consequently, they do not outperform our unified prompts.
> > Even worse, they may risk prompt-answer leakage.
> > This further suggests that loose or low-constrained generative prompts weaken the focused reasoning required for AVQA.
> > We have included these analyses in Appendix A.2.3 and Table 9.
> >
> > | Method | Audio QA | Visual QA | Audio-Visual QA | Avg |
> > | :--- | :---: | :---: | :---: | :---: |
> > | *w/o* Prompts | 79.39 | 83.32 | 75.47 | 78.25 |
> > | *w* Translation | 79.45 | 83.77 | 75.59 | 78.44 |
> > | *w* Caption | 79.33 | 84.35 | 75.06 | 78.28 |
> > | *w* Generative Prompts | 79.21 | **84.56** | 75.16 | 78.37 |
> > | QCR (ours) | **80.63** | 84.17 | **75.98** | **78.98** |
> >
> > [1]. Hurst, A., Lerer, A., Goucher, A.P., Perelman, A., Ramesh, A., Clark, A., Ostrow, A.J., Welihinda, A., Hayes, A., Radford, A. and Madry, A., 2024. Gpt-4o system card. arXiv preprint arXiv:2410.21276.
> >
> > **Q5. Some writing suggestions include avoiding widowed words at the end of paragraphs and adding references for the comparative methods listed in the tables.**
> >
> > A: Thank you for bringing this to our attention.
> > We have addressed these points accordingly.

---

> > > ### Comment · Reviewer_92oA · 2025-11-28
> > >
> > > Thank you very much for the authors’ detailed response, which addressed the vast majority of my concerns. I will maintain my positive review score.

---

> > > > ### Author Response · Authors · 2025-11-28
> > > >
> > > > Dear Reviewer 92oA,
> > > >
> > > > We sincerely appreciate your supportive assessment and are glad that our responses have addressed your concerns. If any additional questions arise, we would be more than happy to provide further clarification.
> > > > Your constructive feedback has been invaluable in strengthening our work!
> > > >
> > > > Best regards,
> > > >
> > > > The Authors

---

### Author Response · Authors · 2025-11-20
**General Response**

Dear Reviewers and ACs,

Thank you so much for your time and efforts in assessing our paper!
We feel very encouraged that the reviewers find our work well-motivated, novel, and intuitive.
The insightful comments are very helpful in improving our work, and we have responded to the concerns point by point.

According to our understanding, there are three common concerns from reviewers:

- Effectiveness of frequency-aware cues. We now provide clearer theoretical motivation, more experiments, and deeper empirical analysis demonstrating how frequency-domain cues contribute across question types.

- Use and construction of prompts. We clarify the design of our unified prompt set, explain how the keywords were derived, and compare our approach against three alternative prompting strategies.

- Comparison with MLLMs. We have conducted new experiments evaluating QSTar against several recent MLLMs and provided a thorough discussion of the performance.


We have taken all the suggestions carefully and updated our previous version.
In the revised manuscript, we have made the following main changes and highlighted them in blue:

- Expanded explanation of frequency-based modeling advantages in Section 1 and 3.3.

- Improved discussion of related works in Section 1, 2, and Appendix A.1.

- Added new comparison and analysis of MLLMs in Section 4.3.

- Added computational cost analysis in Section 4.5.

- Expanded explanation of prompt construction and fairness in Appendix A.2.

- Added new ablation study on the audio feature usage in Appendix A.2.

- Improved the overall presentation of the paper.

We are happy to discuss with you further if you still have other concerns. Thank you very much again!

Best regards,

Paper 7188 Authors

---

### Comment · Area_Chair_38L7 · 2025-11-24

Dear reviewers,

Thank you for your dedicated service as reviewers. Your efforts are critical to the success of our conference, and we deeply appreciate your time and expertise.

This paper has received reviews from reviewers but some have not provided a response to the author rebuttal. Given the limited time we have for author-reviewer discussions, we kindly ask you to share your post-rebuttal feedback to help clarify your perspective and aid the decision-making process.

Your input is invaluable in ensuring a fair and thorough review process.

Best,
AC

---

### Author Response · Authors · 2025-12-01
**Summary of Rebuttal Progress**

Dear ACs,

We sincerely appreciate your time and effort in evaluating our manuscript under the recent exceptional circumstances.
To support your assessment, we provide below a concise summary of the reviews, our key changes and improvements, and the follow-up exchanges with reviewers during the rebuttal period.

1. Strengths acknowledged by the reviewers.

Across the initial reviews, **multiple strengths of the submission were consistently recognized**, including:

- "clear clarity" (92oA, ttss) and "well-structured" (92oA, VpPp)

- "well-motivated" (92oA, ttss) and "interesting and meaningful" (VpPp)

- "clear novelty" (92oA, BoFL)

- "confirmed effectiveness" (92oA, VpPp, BoFL, ttss) and "strong empirical performance" (BoFL)

We are grateful for this positive and encouraging assessment from all reviewers.

2. Concerns raised and our major revisions.

We carefully addressed all reviewer comments.
All substantial changes in the manuscript are **highlighted in blue**, and detailed explanations are provided in our point-by-point responses.

Across the reviews, three concerns were most commonly shared and primarily required further clarification and empirical validation:

- Effectiveness of frequency-aware cues in AVQA. We expanded theoretical motivation, added additional ablations, and provided deeper empirical analysis demonstrating the contribution of frequency-domain modeling.

- Construction of the prompts in QCR and comparison with alternatives. We thoroughly clarified how the prompts are derived, added additional ablations, and compared our approach with three alternative prompting strategies (translation-based, caption-based, and constrained generative prompts).

- Comparison with recent MLLMs. We added a comprehensive evaluation against four representative MLLMs across all question types and provided detailed analysis of the performance differences.

In addition, reviewers requested greater clarity on several components, deeper discussion of computational cost, and improved presentation. All of these have been addressed in the revised paper.
**A complete list of changes** is also summarized in our general response to reviewers and ACs before.


3. Reviewer engagement and recognition during the rebuttal.

**Before the leak, three reviewers actively participated in the rebuttal discussion, and two of them updated their ratings upward from 4 to 6.**
The last reviewer later joined the discussion and also confirmed that the main concerns were addressed.
In more detail:

- **Reviewer BoFL responded on Nov 24**, acknowledged that the main concerns were addressed, and **raised the rating from 4 to 6**.
In the context of the review and score rollback, we emphasize that the authoritative statement provided by BoFL is "After reading the rebuttals and revisions, I believe the author has addressed my main concerns. I am raising my rating from 4 to 6."

- **Reviewer ttss engaged on Nov 26** and requested additional analysis on audio feature extraction. After our extended experiments and explanations, the reviewer indicated that all concerns had been resolved and subsequently **revised the final rating to 6**.

- **Reviewer VpPp confirmed on Nov 26** that most concerns were addressed and maintained a positive score of 6.

- **Reviewer 92oA joined on Nov 28**, stated that our responses addressed the main concerns, and maintained the positive score of 6.

We are grateful that **all reviewers expressed satisfaction with our detailed responses and acknowledged that the majority of their concerns had been fully addressed**.
As evidence of the resolution achieved during the rebuttals, all reviewers concluded the open discussion period with scores of **(6, 6, 6, 6)**, positioning the paper above the acceptance threshold.


We hope this summary clearly reflects how the work evolved through the rebuttal process and how reviewers' concerns were addressed and acknowledged.
For full details, we refer you to our point-by-point responses.
We welcome any additional questions and sincerely appreciate your (both the initial and newly assigned ACs) time and thoughtful handling of this paper.

Best regards,

Paper 7188 Authors

---

### Meta-Review · Area_Chair_aeZq · 2026-01-07

**Summary:**

The reviewers generally agreed that the paper proposes a well-motivated approach to music audio–visual question answering, which incorporates frequency-domain audio cues and query-guided multimodal interaction. The major concerns raised in the initial reviews included:
(1) the motivation is not clearly supported by any evidence,
(2) the necessity and efficiency of introducing frequency-domain modeling,
(3) the generality of the approach beyond the MUSIC-AVQA dataset and the scalability of the prompt design, and
(4) the lack of comparison with recent multimodal large language models (MLLMs).

**Reviewer Concerns:**

***Concerns addressed by the rebuttal:***

Motivation: The authors provided concrete analyses and comparisons clarifying how prior AVQA methods are largely vision-centric or integrate audio/text late, and demonstrated the benefits of early query-guided alignment and frequency-aware modeling through new ablations.

Justification of frequency-domain modeling (AST): The rebuttal clarified the complementary roles of AST and VGGish, added backbone ablations (including AST-only and AST+VGGish variants).

Efficiency and complexity: Additional parameter/FLOPs analysis was provided.

Prompt design and fairness: The authors explained the rationale for unified, generic prompts, provided comparisons with translation-based, caption-based, and generative prompts, and demonstrated that alternative prompting strategies do not outperform the proposed design.

Comparison with MLLMs: New experiments comparing against both zero-shot proprietary MLLMs (e.g., GPT-4o, Qwen2.5-Omni, Ming-Omni) and a fine-tuned open-source MLLM (VideoLLaMA2) were added, showing clear performance gaps in favor of the proposed method on AVQA tasks.

Depth of analysis: The rebuttal added question-type–level analysis, rare-instrument breakdowns, qualitative examples, and additional discussion clarifying where frequency cues are most impactful.

***Remaining or partially outstanding concerns:***

The scope remains primarily focused on music-centric AVQA, and while additional AVQA results are included, generalization beyond music scenes is still more limited than Reviewer BoFL initially hoped.

Model novelty is incremental in the sense that the framework builds on standard attention mechanisms; the contribution lies more in problem-driven architectural composition and frequency-aware interaction design than in introducing a fundamentally new network module.

**Reviewer Scores:**

Reviewer 92oA (initial score: 6):
Confirmed that the rebuttal addressed the majority of concerns and maintained a positive rating.

Reviewer VpPp (initial score: 6):
After the additional clarifications, indicated that most concerns were resolved and maintained the positive score.

Reviewer BoFL (initial score: 4 → 6):
Initially raised substantial concerns regarding motivation, dataset scope, prompt fairness, and the absence of MLLM comparisons. These issues were directly addressed in the rebuttal, and the reviewer subsequently increased the score in discussions.

Reviewer ttss (initial score: 4 → 6):
Initially expressed concerns about limited novelty, efficiency, and missing MLLM comparisons. The rebuttal added efficiency analysis, audio-backbone ablations, and MLLM evaluations, leading the reviewer to explicitly revise the score upward.

Following the rebuttal and discussion, reviewer assessments converged toward a positive consensus. The AC sees no reason to overturn the reviewers’ evaluations and therefore recommends acceptance as a poster.

---

### Decision · Program_Chairs · 2026-01-26

Accept (Poster)